# PTCL: Pseudo-Label Temporal Curriculum Learning for Label-Limited Dynamic Graph

## Abstract

Dynamic node classification is critical for modeling evolving systems like financial transactions and academic collaborations. In such systems, dynamically capturing node information changes is critical for dynamic node classification, which usually requires all labels at every timestamp. However, it is difficult to collect all dynamic labels in real-world scenarios due to high annotation costs and label uncertainty (e.g., ambiguous or delayed labels in fraud detection). In contrast, final timestamp labels are easier to obtain as they rely on complete temporal patterns and are usually maintained as a unique label for each user in many open platforms, without tracking the history data. To bridge this gap, we propose a pioneering method PTCL (**P**seudo-label **T**emporal **C**urriculum **L**earning), combining the variational EM (Expectation Maximization) framework with a novel Temporal Curriculum Learning strategy to effectively leverage both final timestamp labels and pseudo-labels. We also contribute a new academic dataset CoOAG (**Co**-authorship graph derived from **O**pen **A**cademic **G**raph), capturing long-range research interest in dynamic graph. Experiments across real-world scenarios demonstrate PTCL's consistent superiority over other methods adapted to this task. Beyond methodology, we propose a unified framework FLiD (**F**ramework for **L**abel-**Li**mited **D**ynamic Node Classification), consisting of a complete preparation workflow, training pipeline, and evaluation standards, and supporting various models and datasets. Code details can be found in supplementary materials.

## 1 Introduction

Graph-structured data is widespread in domains such as social networks (Ying et al., 2018; Newman et al., 2002; Feng et al., 2022), biological systems (Zitnik et al., 2018; Li et al., 2022a), financial transactions (Huang et al., 2022; Li & Yang, 2023), and academic collaborations (Hu et al., 2021; Zhou et al., 2022). Graphs effectively capture relationships between entities, enabling powerful modeling of complex systems (Bronstein et al., 2017). A key task is node classification, which assigns labels to nodes based on features and structure (Xiao et al., 2022; Bhagat et al., 2011; Rong et al., 2019), and has been extensively studied in static graphs. However, many real-world graphs are dynamic, with evolving nodes, edges, and labels over time (Rossi et al., 2020; Kumar et al., 2019; Xu et al., 2020). For instance, authors may shift research areas (Jia et al., 2017) or users may change behaviors in transaction networks (Huang et al., 2022), highlighting the need for dynamic node classification that accounts for temporal label changes.

Ideally, training on a complete dynamic trajectory yields a strong classifier. However, dynamic node classification is challenged by the high cost of acquiring dynamic labels, requiring continuous monitoring, manual annotation, and coping with uncertainty (e.g., delayed or ambiguous fraud labels). Existing dynamic datasets often provide weak or rarely changing labels (Kumar et al., 2019), limiting their ability to reflect evolving node behavior. While labeling nodes at every timestamp is difficult, obtaining a final label (e.g., fraud status) at the end of a period is more feasible, as illustrated in Figure 1 (Huang et al., 2022). Many platforms, such as OAG (Open Academic Graph) (Sinha et al.; Zhang et al., b;a; Tang et al.), also offer only final static labels (e.g., fixed research interests). This motivates our core task: **label-limited dynamic node classification**. The goal is to classify nodes in dynamic graphs using limited label information, especially final timestamp labels, which implicitly summarize long-term behavior. Effectively leveraging unlabeled historical data thus requires robust modeling of temporal dynamics.

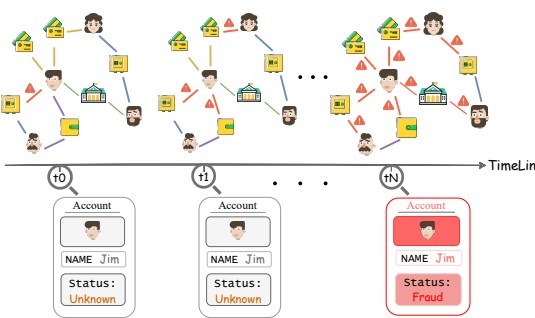

Figure 1: A representation of a dynamic financial system. The graph models entities such as users, payment cards, and financial institutions as nodes, with edges denoting transactional relationships. Over time, user behavior evolves through sequences of transactions, forming dynamic interaction patterns. Some users' labels (e.g., account status) may eventually be revealed as fraudulent based on long-term behavioral signals.

Building on the foundation of SSL (Semi-Supervised Learning) (Yang et al., 2022; Zhou et al., 2020; Zhu, 2005; Learning, 2006; Chapelle et al., 2006; Van Engelen & Hoos, 2020) and pseudo-labeling (Lee et al., 2013; Kage et al., 2024), we propose `PTCL` (**P**seudo-label **T**emporal **C**urriculum **L**earning), which consists of a dynamic graph encoder (backbone) and a label predictor (decoder) (Yu et al., 2023; Rossi et al., 2020). Inspired by the variational EM framework (Qu, 2024; Qu et al., 2019; Zhao et al., 2022; Neal & Hinton, 1998; Dempster et al., 1977), we decouple their optimization: the decoder is trained solely on final timestamp labels to ensure label fidelity, then used to generate pseudo-labels for earlier timestamps. These pseudo-labels, combined with the final labels, guide the backbone training via a weighted loss, allowing it to learn temporal label dynamics effectively. To mitigate the impact of low-quality pseudo-labels in early EM iterations, we introduce a **Temporal Curriculum Learning** strategy inspired by **Curriculum Learning** (Bengio et al., 2009; Wang et al., 2021b; Soviany et al., 2021). Following the easy-to-hard principle, we assign higher weights to pseudo-labels near the final timestamps—where predictions are more reliable—at early stages of training. Over time, the model gradually incorporates earlier, harder timestamps, enabling progressive learning of temporal dynamics.

Extensive Experiments show that `PTCL` consistently outperforms other methods adapted to this task across multiple datasets, validating its effectiveness in capturing the temporal evolution of nodes. Additionally, we conduct a series of studies to verify the contribution of each design of `PTCL`.

To sum up, our contributions are as follows:

- **Pioneering Study**: To the best of our knowledge, this is a pioneering study to systematically investigate the problem of label-limited dynamic node classification. We formalize the task, identify its unique challenges, and propose a comprehensive method to address them.
- **Novel Method**: We introduce a new method `PTCL`, which captures the dynamic nature of nodes with limited labels, advancing dynamic graph study by constructing highly varying history information. Various experiments demonstrate the effectiveness of `PTCL` and the necessity of each design.
- **New Dataset**: We contribute a new dataset, CoOAG, which is derived from academic collaboration networks and designed for dynamic graph learning. It captures the dynamic nature of research interests, providing a rich testbed for evaluating `PTCL`.
- **Unified Framework**: We propose a unified framework, FLiD (**F**ramework for **L**abel-**Li**mited **D**ynamic Node Classification), for our task, which includes a complete preparation workflow, a training pipeline, and evaluation protocols. Our framework supports various backbones and datasets, offering a flexible and extensible solution.

## 2 PROBLEM FORMULATION

A **dynamic graph with dynamic labels** can be mathematically represented as a sequence of chronologically ordered events: $\mathcal{G} = \{x(t_i)\} = \{(u_i, v_i, t_i)\}$, where $0 \leq t_1 \leq t_2 \leq \cdots$. Each event $x(t_i)$ describes an interaction between a source node $u_i \in \mathcal{V}$ and a destination node $v_i \in \mathcal{V}$ at time $t_i$. And $y_{u_i}^{t_i}, y_{v_i}^{t_i} \in \mathcal{Y}$ are their respective labels at $t_i$. $\mathcal{V}$ denotes the set of all nodes, and $\mathcal{Y}$ is the class set of all nodes. For each node $u \in \mathcal{V}$, $\mathcal{T}_u = \{t_i \mid u = u_i \text{ or } u = v_i \text{ in } x(t_i) \in \mathcal{G}\}$ is the set of all timestamps at which $u$ participates in any event in $\mathcal{G}$. The last occurrence time $T_u = \max \mathcal{T}_u$ is the most recent timestamp in $\mathcal{T}_u$. We further define $\mathcal{Y}_F = \{y_u^{T_u} \mid u \in \mathcal{V}\}$, the set of ground-truth labels at the final timestamps, and $\mathcal{Y}_E = \{y_u^t \mid u \in \mathcal{V}, t \in \mathcal{T}_u \setminus \{T_u\}\}$, the set of labels at non-last timestamps

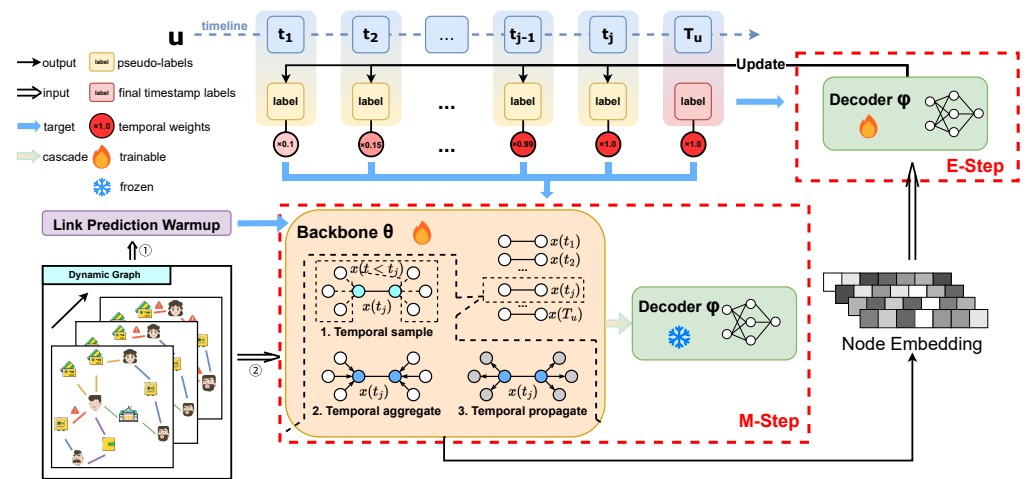

Figure 2: Overview of our proposed method. PTCL consists of a Variational EM process with a dynamic graph backbone and a decoder. During the warmup phase, the dynamic graph backbone is trained on a link prediction task, where the dynamic graph structure serves as the target. After warmup, in each M-step, the backbone receives final timestamp labels, pseudo-labels, and the dynamic graph structure as input, while the decoder is trained in the E-step to refine pseudo-labels. Additionally, the Temporal Curriculum Learning strategy prioritizes pseudo-labels based on their temporal proximity to the final timestamp labels, ensuring higher-quality training.

of every node. In most cases, $|\mathcal{Y}_E| \gg |\mathcal{Y}_F|$. In our research scenario, $\mathcal{Y}_F$ are known, whereas $\mathcal{Y}_E$ are considered unknown due to data collection constraints. Following the training-evaluation paradigm, we determine a boundary time $T_B$ to separate the training and evaluation datasets. The final timestamp label set $\mathcal{Y}_F$ is then divided into two subsets: $\mathcal{Y}_{F,B} = \{y_u^{T_u} \mid u \in \mathcal{V}, T_u \leq T_B\}$ consisting of labels for nodes whose final timestamps are before $T_B$, and $\mathcal{Y}_{F,A} = \{y_u^{T_u} \mid u \in \mathcal{V}, T_u > T_B\}$ containing labels for nodes whose final timestamps is after $T_B$. Similarly, $\mathcal{Y}_E$ is partitioned into $\mathcal{Y}_{E,B} = \{y_u^t \mid u \in \mathcal{V}, t \in \mathcal{T}_u \setminus \{T_u\}, t \leq T_B\}$ and $\mathcal{Y}_{E,A} = \{y_u^t \mid u \in \mathcal{V}, t \in \mathcal{T}_u \setminus \{T_u\}, t > T_B\}$, representing labels whose corresponding timestamps are before and after $T_B$, respectively.

The dynamic graph backbone generates node embeddings $\mathbf{h}_u^t$ for each node $u$ at each timestamp $t \in \mathcal{T}_u$. The backbone takes the node features as input $\mathbf{n}_u \in \mathbb{R}^{d_N}$ and edge features $\mathbf{e}_{u,v}^t \in \mathbb{R}^{d_E}$. If the graph is non-attributed, we assume $\mathbf{n}_u = \mathbf{0}$ and $\mathbf{e}_{u,v}^t = \mathbf{0}$ for all nodes and edges, respectively.

Given a dynamic graph $\mathcal{G}$ with dynamic labels and $|\mathcal{Y}_E| \gg |\mathcal{Y}_F|$, our task **label-limited dynamic node classification** aims to maximize $\log p(\mathcal{Y}_{F,B}|\mathcal{G})$. Specifically, our goal is to learn a model that can finally accurately predict $\mathcal{Y}_F$.

## 3 METHODOLOGY

In this section, we present PTCL for label-limited dynamic label learning. As shown in Figure 2, PTCL combines the variational EM framework with a Temporal Curriculum Learning strategy to effectively leverage both final timestamp labels and pseudo-labels in a dynamic graph setting.

### 3.1 VARIATIONAL EM FRAMEWORK

Following previous work (Zhao et al., 2022; Qu et al., 2019), we adopt the variational EM framework (Dempster et al., 1977; Neal & Hinton, 1998) to maximize $\log p(\mathcal{Y}_{F,B}|\mathcal{G})$.

### 3.1.1 EVIDENCE LOWER BOUND (ELBO)

To handle the unknown labels $\mathcal{Y}_{E,B}$, instead of directly optimizing $\log p_\theta(\mathcal{Y}_{F,B}|\mathcal{G})$, we maximize the evidence lower bound (ELBO) of the log-likelihood function:

$$\log p(\mathcal{Y}_{F,B}|\mathcal{G}) \geq \mathbb{E}_{q_\phi(\mathcal{Y}_{E,B}|\mathcal{G})}[\log p_\theta(\mathcal{Y}_{F,B}, \mathcal{Y}_{E,B}|\mathcal{G}) - \log q_\phi(\mathcal{Y}_{E,B}|\mathcal{G})], \tag{1}$$

where $p_\theta(\mathcal{Y}_{F,B}, \mathcal{Y}_{E,B}|\mathcal{G})$ is the joint distribution of observed and unknown labels, modeled by the dynamic graph backbone with parameters $\theta$. $q_\phi(\mathcal{Y}_{E,B}|\mathcal{G})$ is the variational distribution approximating the true posterior distribution $p_\theta(\mathcal{Y}_{E,B}|\mathcal{Y}_{F,B}, \mathcal{G})$, modeled by the decoder with parameters $\phi$.

To facilitate optimization, we follow mean field assumption (Getoor et al., 2001), which yields the following factorization:

$$q_\phi(\mathcal{Y}_{E,B}|\mathcal{G}) = \prod_{u \in \mathcal{V}} \prod_{\substack{t \in \mathcal{T}_u \setminus \{T_u\} \\ t \leq T_B}} q_\phi(y_u^t|\mathbf{h}_u^t), \tag{2}$$

where $q_\phi(y_u^t|\mathbf{h}_u^t)$ is the label distribution predicted by the decoder.

The ELBO can be optimized by alternating between the E-step and the M-step.

### 3.1.2 E-STEP

In the E-step, we use the wake-sleep algorithm (Hinton et al., 1995), following (Zhao et al., 2022). We fix the dynamic graph backbone $\theta$ and optimize the decoder $\phi$ to minimize the KL divergence between the true posterior distribution $p_\theta(\mathcal{Y}_{E,B}|\mathcal{G}, \mathcal{Y}_{F,B})$ and the variational distribution $q_\phi(\mathcal{Y}_{E,B}|\mathcal{G})$. The objective function for the decoder is:

$$\hat{\mathcal{O}}_\phi = \sum_{u \in \mathcal{V}} \sum_{\substack{t \in \mathcal{T}_u \setminus \{T_u\} \\ t \leq T_B}} \mathbb{E}_{p_\theta(y_u^t|\mathcal{G}, \mathcal{Y}_{F,B})} \left[ \log q_\phi(y_u^t|\mathbf{h}_u^t) \right], \tag{3}$$

Following (Zhao et al., 2022), we use the pseudo-labels $\hat{\mathcal{Y}}_{E,B}$ generated by the decoder to approximate the distribution $p_\theta(y_u^t|\mathcal{G}, \mathcal{Y}_{F,B})$:

$$p_\theta(y_u^t|\mathcal{G}, \mathcal{Y}_{F,B}) \approx p_\theta(y_u^t|\mathcal{G}, \mathcal{Y}_{F,B}, \hat{\mathcal{Y}}_{E,B} \setminus \{\hat{y}_u^t\}), \tag{4}$$

then the objective function of the decoder changes to:

$$\hat{\mathcal{O}}_\phi = \alpha \sum_{u \in \mathcal{V}} \sum_{\substack{t \in \mathcal{T}_u \setminus \{T_u\} \\ t \leq T_B}} \mathbb{E}_{p_\theta(y_u^t|\mathcal{G}, \mathcal{Y}_{F,B}, \hat{\mathcal{Y}}_{E,B} \setminus \{\hat{y}_u^t\})} \left[ \log q_\phi(y_u^t|\mathbf{h}_u^t) \right] + (1 - \alpha) \sum_{\substack{u \in \mathcal{V} \\ T_u \leq T_B}} \log q_\phi(y_u^{T_u}|\mathbf{h}_u^{T_u}), \tag{5}$$

where $\alpha$ is a hyperparameter that balances the weight of pseudo-labels and final timestamp labels.

But in practice, as shown in Section 4.2, we find that setting $\alpha$ to 0 yields the best performance, which means we train the decoder only with final timestamp labels. Therefore, the final objective function for the decoder is:

$$\mathcal{O}_\phi = \sum_{\substack{u \in \mathcal{V} \\ T_u \leq T_B}} \log q_\phi(y_u^{T_u}|\mathbf{h}_u^{T_u}), \tag{6}$$

### 3.1.3 M-STEP

In the M-step, following the previous work (Zhao et al., 2022; Qu et al., 2019), we aim to maximize the following pseudo-likelihood (Besag, 1975). Specifically, we fix the decoder $\phi$ and optimize the dynamic graph backbone $\theta$ using both the final timestamp labels $\mathcal{Y}_{F,B}$ and the pseudo-labels $\hat{\mathcal{Y}}_{E,B}$ generated in the E-step. The objective is to maximize the pseudo-likelihood:

$$\hat{\mathcal{O}}_\theta = \beta \sum_{u \in \mathcal{V}} \sum_{\substack{t \in \mathcal{T}_u \setminus \{T_u\} \\ t \leq T_B}} \log p_\theta(y_u^t|\mathcal{G}, \mathcal{Y}_{F,B}, \hat{\mathcal{Y}}_{E,B} \setminus \{\hat{y}_u^t\})$$

$$+ (1 - \beta) \sum_{\substack{u \in \mathcal{V} \\ T_u \leq T_B}} \log p_\theta(y_u^{T_u}|\mathcal{G}, \mathcal{Y}_{F,B} \setminus \{y_u^{T_u}\}, \hat{\mathcal{Y}}_{E,B}), \tag{7}$$

where $\beta$ is a hyperparameter that balance the weight of pseudo-labels and final timestamp labels. Since the backbone can only generate embeddings, we cascade the backbone and the fixed decoder, using final timestamp labels and pseudo-labels generated by the decoder to train the backbone.

## 3.2 Temporal Curriculum Learning

In the M-step, the backbone is trained on both final labels $\mathcal{Y}_{F,B}$ and pseudo-labels $\hat{\mathcal{Y}}_{E,B}$. To mitigate noise from unreliable pseudo-labels at earlier timestamps, we introduce a **Temporal Curriculum Learning** strategy that dynamically adjusts pseudo-label weights based on their temporal proximity to the final timestamp and the EM iteration $\tau$.

Specifically, we design a weighting mechanism for pseudo-labels based on their temporal order relative to the final timestamp $T_u$. Specifically, for each node $u \in \mathcal{V}$, timestamp $t \in \mathcal{T}_u$ in $\tau$-th iteration, we define a weight $w_u^{t,\tau}$ as follows:

$$
w_u^{t,\tau} = f_{\text{TW}}(t, u, \tau, T_u, \gamma) = \begin{cases} 1, & \text{if } d_u^t \leq \tau, \\ \exp\left(-\gamma \cdot (d_u^t - \tau)\right), & \text{if } d_u^t > \tau, \end{cases} \tag{8}
$$

$$
d_u^t = |\{t' \in \mathcal{T}_u \mid t' > t\}|, \tag{9}
$$

where $d_u^t$ is the discrete temporal distance between of timestamp $t$ and $T_u$ in $\mathcal{T}_u$. $\gamma > 0$ is a hyperparameter that controls the rate of Temporal Curriculum Learning decay. $w_u^{t,\tau}$ dynamically adjusts the importance of pseudo-labels during training. If $d_u^t \leq \tau$, the timestamp $t$ is considered close to the final timestamp $T_u$, and the pseudo-label is assigned a weight of 1, indicating high confidence in its quality. And if $d_u^t > \tau$, the timestamp $t$ is considered far from $T_u$, and the pseudo-label weight decays exponentially with the distance $\tau - d_u^t$, reducing its influence on the training process.

With the pseudo-label temporal weights $w_u^{t,\tau}$, the objective function for the M-step is modified as follows:

$$
\begin{aligned}
\mathcal{O}_\theta = \beta \sum_{u \in \mathcal{V}} \sum_{\substack{t \in \mathcal{T}_u \setminus \{T_u\} \\ t \leq T_B}} & w_u^{t,\tau} \log p_\theta(y_u^t | \mathcal{G}, \mathcal{Y}_{F,B}, \hat{\mathcal{Y}}_{E,B} \setminus \{\hat{y}_u^t\}) \\
& + (1 - \beta) \sum_{\substack{u \in \mathcal{V} \\ T_u \leq T_B}} \log p_\theta(y_u^{T_u} | \mathcal{G}, \mathcal{Y}_{F,B} \setminus \{y_u^{T_u}\}, \hat{\mathcal{Y}}_{E,B}).
\end{aligned} \tag{10}
$$

By incorporating $w_u^{t,\tau}$, the backbone is trained to prioritize high-quality pseudo-labels.

## 3.3 Learning and Optimization

Since the initial parameters of the EM algorithm are crucial for its performance (Dy & Brodley, 2004; Kwedlo, 2015), we first warm up the backbone by training it on a link prediction task. Then we proceed with the variational EM algorithm, which alternates between the E-step and the M-step. Finally, we use the decoder to predict $\mathcal{Y}_{F,A}$. The complete algorithm is summarized in Appendix.C.

## 4 Experiments

Our experiments are designed to address the following key research questions (RQs):

**RQ1**: How does `PTCL` perform compared to other baselines when evaluated on the final timestamp labels? **RQ2**: Does pseudo-labels generated by `PTCL` improve performance by capturing the dynamic information of nodes? **RQ3**: Does the Temporal Curriculum Learning strategy improve performance? **RQ4**: Is `PTCL` stable and computationally efficient? More experiments can be found in Appendix.F.

### 4.1 Experiment Settings

#### 4.1.1 Datasets

We evaluate `PTCL` on four datasets:

Figure 3: Architectures of baselines and `PTCL`. 'BB': backbone, 'Dec': decoder.

- **Wikipedia** (Kumar et al., 2019) and **Reddit** (Kumar et al., 2019): Bipartite interaction graphs capturing user activities over one month. These datasets feature binary classification tasks predicting banned users due to policy violations (Wikipedia) or community guideline violations (Reddit). Labels evolve dynamically as user behavior changes over time.
- **Dsub**: A subgraph of Dgraph (Huang et al., 2022) representing a financial transaction network. The binary classification task focuses on fraud detection—identifying users who ultimately fail to repay loans. Many nodes in this dataset represent background users that lack sufficient information for labeling but are retained to maintain graph connectivity.
- **CoOAG**: Our novel academic co-authorship graph derived from OAG (Open Academic Graph (Sinha et al.; Zhang et al., b;a; Tang et al.)). Nodes represent authors and edges denote co-authorship on AI conference papers. Node labels reflect evolving research interests across five fields: Computer Vision (CV), Natural Language Processing (NLP), Robotics (ROB), Data Mining/Web Search (DM/WS), and other AI/ML fields. Crucially, 43.2% of nodes change labels at least once, making it particularly valuable for studying label dynamics. Construction procedure of CoOAG is detailed in Appendix D.2.2, while comprehensive details for all datasets are provided in Appendix D.

For the binary classification tasks (Wikipedia, Reddit, Dsub), we report AUC (excluding background nodes in Dsub); for the multi-class CoOAG, we use ACC. To reflect real-world scenarios where only final labels are available, we adopt a timestamp-based split and evaluate solely on final labels ($\mathcal{Y}_{F,A}$). Nodes are split into train/val/test sets at a 7:1.5:1.5 ratio based on label distributions. All results are averaged over five random seeds. Implementation and hyperparameters are in Appendix J and K.

### 4.1.2 BASELINES

We compare `PTCL` with several different methods that can be adapted to our task. And specifically, we use five different dynamic backbones as backbones: TGAT (Xu et al., 2020), GraphMixer (Cong et al., 2023), TCL (Wang et al., 2021a), TGN (Rossi et al., 2020), and DyGFormer (Yu et al., 2023), and apply a simple 3-layer MLP as the decoder. More backbone details are in Appendix.E. As shown in Figure 3, the baselines are designed to cover a range of approaches:

- **CFT (Copy-Final Timestamp labels)**: A naive baseline that simply copies the final timestamp labels ($\mathcal{Y}_{F,B}$) to earlier timestamps as approximations of dynamic labels ($\mathcal{Y}_{E,B}$) for training.
- **DLS (Dynamic Label Supervision)**: A baseline that performs supervised training directly using the dynamic labels provided by the dataset (e.g., Wikipedia, Reddit), where available (Yu et al., 2023; Rossi et al., 2020).
- **NPL (Naive Pseudo-Labels)**: A variant of `PTCL` that uses pseudo-labels but jointly optimizes the backbone and decoder without EM optimization.
- **PTCL-2D (`PTCL` with 2 Decoders)**: A variant of `PTCL` that uses two decoders: one decoder is trained exclusively on the final timestamp labels (E-step), generating pseudo-labels, while the other decoder is jointly optimized with the backbone on weighted pseudo-labels and final timestamp labels (M-step). The final embeddings are provided by the backbone for the E-step training.
- **SEM (Standard EM)**: A variant of `PTCL` where both the E-step and M-step are trained on the weighted loss of pseudo-labels and final timestamp labels (Zhao et al., 2022), while other components remain the same as `PTCL`. In this way, E-step uses Eq. (5) as the objective function instead of Eq. (6).

Table 1: Performance comparison across datasets (Wikipedia, Reddit, Dsub, CoOAG). We run all experiments with five random seeds to ensure a consistent evaluation and report the average performance as well as standard deviation in parentheses. **Bold** indicates the best performance, underline the second best. Dsub and CoOAG datasets can not apply the DLS method due to a lack of dynamic labels. TGN runs out of memory on Dsub due to its high space cost.

| Backbone | Method | Wikipedia | Reddit | Dsub | CoOAG |
|---|---|---|---|---|---|
| | | AUC | AUC | AUC | ACC |
| TGAT | CFT | 77.43 (± 3.01) | 82.68 (± 0.06) | 62.32 (± 1.27) | 86.28 (± 0.18) |
| | DLS | 79.56 (± 2.55) | 78.66 (± 0.04) | – | - |
| | NPL | 78.52 (± 2.28) | 80.72 (± 2.65) | 61.71 (± 2.21) | 87.67 (± 0.61) |
| | PTCL-2D | 78.20 (± 6.83) | 85.84 (± 6.09) | 60.76 (± 2.91) | 88.37 (± 0.16) |
| | SEM | 81.09 (± 3.62) | 86.27 (± 5.99) | 64.34 (± 0.99) | 88.38 (± 0.38) |
| | **Ours** | **85.52** (± 3.29) | **87.31** (± 6.50) | **65.07** (± 1.57) | **89.05** (± 0.63) |
| TCL | CFT | 76.27 (± 4.68) | 84.48 (± 5.53) | 62.60 (± 1.26) | 86.12 (± 1.11) |
| | DLS | 80.55 (± 1.93) | 82.85 (± 0.38) | – | – |
| | NPL | 77.71 (± 5.66) | 84.20 (± 3.86) | 63.59 (± 3.09) | 87.59 (± 0.26) |
| | PTCL-2D | 76.68 (± 2.86) | 86.98 (± 3.34) | 60.64 (± 1.41) | 87.94 (± 0.30) |
| | SEM | 81.02 (± 2.82) | 87.56 (± 1.56) | 65.11 (± 1.26) | 87.90 (± 0.18) |
| | **Ours** | **82.27** (± 4.62) | **89.41** (± 3.32) | **66.80** (± 2.45) | **88.24** (± 0.26) |
| TGN | CFT | 80.68 (± 2.02) | **89.69** (± 2.07) | OOM | 83.65 (± 0.63) |
| | DLS | 78.48 (± 1.60) | 80.92 (± 4.99) | – | – |
| | NPL | 87.58 (± 2.14) | 84.22 (± 2.48) | OOM | 86.13 (± 0.31) |
| | PTCL-2D | 86.59 (± 3.01) | 86.76 (± 3.89) | OOM | 86.23 (± 0.66) |
| | SEM | 86.34 (± 2.66) | 82.61 (± 3.07) | OOM | 86.07 (± 0.66) |
| | **Ours** | **87.97** (± 2.90) | 84.32 (± 2.07) | OOM | **86.71** (± 0.66) |
| GraphMixer | CFT | 76.60 (± 2.00) | 66.11 (± 6.04) | 62.78 (± 1.90) | 85.63 (± 0.14) |
| | DLS | 80.70 (± 4.00) | 61.97 (± 7.36) | – | – |
| | NPL | 80.86 (± 1.62) | 71.72 (± 6.48) | 67.14 (± 1.68) | 86.98 (± 0.61) |
| | PTCL-2D | 81.41 (± 4.25) | 66.86 (± 11.14) | 62.33 (± 1.35) | 87.51 (± 0.51) |
| | SEM | 83.33 (± 1.45) | 68.65 (± 3.70) | 69.23 (± 1.92) | 88.07 (± 0.30) |
| | **Ours** | **84.09** (± 0.95) | **71.93** (± 7.94) | **69.76** (± 1.54) | **88.26** (± 0.38) |
| DyGFormer | CFT | 64.76 (± 9.21) | 67.14 (± 8.04) | 68.48 (± 1.46) | 85.27 (± 0.83) |
| | DLS | 71.95 (± 2.29) | 64.63 (± 4.90) | – | – |
| | NPL | 73.85 (± 5.44) | 67.44 (± 3.47) | 70.31 (± 1.11) | 86.16 (± 0.38) |
| | PTCL-2D | 66.48 (± 6.76) | 71.14 (± 6.27) | 69.11 (± 2.96) | 86.04 (± 0.30) |
| | SEM | 70.91 (± 8.80) | 71.59 (± 4.51) | 69.75 (± 2.47) | 86.07 (± 0.66) |
| | **Ours** | **74.85** (± 3.07) | **75.86** (± 8.04) | **72.39** (± 1.91) | **86.26** (± 0.27) |

### 4.1.3 FLiD: A novel framework

We introduce **FLiD**, a new code framework tailored for scenarios with only final timestamp labels. Existing frameworks such as DyGLib (Yu et al., 2023) and TGL (Zhou et al., 2022) have made important progress in dynamic graph learning: DyGLib unifies models for fully supervised link prediction and node classification, while TGL focuses on scalable training for large dynamic graphs. However, neither framework directly addresses the label-limited dynamic node classification setting considered in this work.

All experiments in this paper are conducted in FLiD, featuring:

- Support for multiple training paradigms, including CFT, DLS, NPL, SEM, PTCL, PTCL-2D;
- Pseudo-labeling enhancements such as Confidence Score Threshold (CST), Entropy of Softmax Trajectory (EST), and Temporal Curriculum Learning;
- A variety of dynamic graph backbones, including TGAT, TGN, GraphMixer, TCL, and DyGFormer;
- A custom data preprocessing and splitting strategy designed for the label-limited setting.

### 4.2 RQ1: Main Results

To evaluate the effectiveness of PTCL, we conduct experiments using five different backbones and compare against several baselines. As shown in Table 1, PTCL consistently improves performance across all datasets and backbones.

Table 2: AUC comparison of different backbones using Dynamic Label Supervised Learning (DLS) and Pseudo-Label Supervised Learning (PLS) on Wikipedia Dataset.

|  | TGAT | TCL | TGN | GraphMixer | DyGFormer |
| --- | --- | --- | --- | --- | --- |
| DLS | 79.56 ($\pm$ 2.55) | 80.55 ($\pm$ 1.93) | 78.48 ($\pm$ 1.60) | 80.70 ($\pm$ 4.00) | 71.95 ($\pm$ 2.29) |
| PLS | 81.11 ($\pm$ 5.12) | 82.19 ($\pm$ 0.64) | 79.33 ($\pm$ 2.58) | 81.02 ($\pm$ 2.94) | 72.62 ($\pm$ 2.07) |

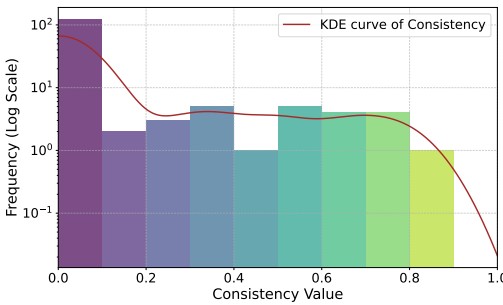

(a) Histogram of pseudo-labels consistency.

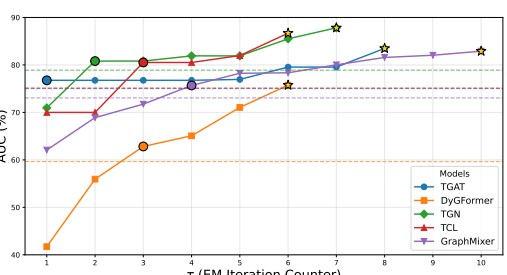

(b) Convergence curves for 5 backbones. Star markers ($\star$) denote peak performance; circled points ($\bullet$) indicate surpassing baselines. Dashed lines show baseline AUC.

Figure 4: (a) Pseudo-label consistency; (b) Convergence of backbones.

- **Effectiveness of Pseudo-Labels.** `PTCL` significantly outperforms both CFT and DLS (up to +11.23% in AUC/ACC), showing that learned pseudo-labels better capture node dynamics than copied or even original dynamic labels (details in Section 4.3).

- **Importance of Our Optimization Strategy.** Compared to NPL, `PTCL` achieves +2.74% improvement on average. Additionally, `PTCL` outperforms SEM, demonstrating that training the decoder exclusively on final timestamp labels ($\alpha = 0$, as described in Section 3.1.2) leads to better results. Although different with the original variaition EM framework, we regard it as a trade-off between strict mathematical adherence to the standard variational EM M-step and engineering effectiveness in dealing with pseudo-labels. These results highlight the benefit of optimizing the decoder solely with final labels, which ensures alignment with ground truth and prevents error propagation from noisy pseudo-labels.

- **Efficiency over `PTCL`-2D.** `PTCL` achieves better performance with lower compute cost than `PTCL`-2D variant, which indicates that a single decoder shared by E-step and M-step promotes more stable and consistent training .

### 4.3 RQ2: PSEUDO-LABEL ANALYSIS

In this section, we evaluate the effectiveness of our pseudo-labels and their ability to capture dynamic patterns through two experiments.

#### 4.3.1 PSEUDO-LABEL SUPERVISION STUDY

To analyze the effectiveness of our pseudo-labels, we conduct the following experiment: We train models from scratch using pseudo-labels generated by our trained model as full supervision labels and compare the results with DLS. As shown in Table 2, models trained with our pseudo-labels consistently outperform those trained with original dynamic labels with an average improvement of 0.96% in AUC.

#### 4.3.2 LABEL CONSISTENCY ANALYSIS

To further investigate the temporal changes of labels and to explain the counter-intuitive result where the Dynamic Label Supervision (DLS) baseline underperforms simple pseudo-labeling methods (Table 1), we analyze the labels' consistency on Wikipedia's positive samples. Consistency is

Table 3: AUC comparison of our Temporal Curriculum Learning with other strategies. CST = Confidence Score Threshold, EST = Entropy of Softmax Trajectory.

| Backbone | Dataset | Naive | CST | EST | Ours |
|---|---|---|---|---|---|
| TGAT | Wikipedia | 82.38 ($\pm$ 4.16) | 79.48 ($\pm$ 2.84) | 81.89 ($\pm$ 4.83) | **85.52** ($\pm$ 3.29) |
| | Dsub | 64.12 ($\pm$ 1.87) | 62.78 ($\pm$ 2.27) | 63.80 ($\pm$ 1.45) | **65.07** ($\pm$ 1.57) |
| TCL | Wikipedia | 81.77 ($\pm$ 1.45) | 78.06 ($\pm$ 2.59) | 80.56 ($\pm$ 5.75) | **82.27** ($\pm$ 4.62) |
| | Dsub | 65.29 ($\pm$ 1.56) | 63.68 ($\pm$ 1.36) | 64.57 ($\pm$ 1.53) | **66.80** ($\pm$ 2.45) |
| TGN | Wikipedia | 86.33 ($\pm$ 3.51) | 83.63 ($\pm$ 1.33) | 86.88 ($\pm$ 1.52) | **87.97** ($\pm$ 2.90) |
| | Dsub | OOM | OOM | OOM | OOM |
| GraphMixer | Wikipedia | 83.34 ($\pm$ 2.22) | 81.59 ($\pm$ 2.54) | 81.13 ($\pm$ 1.44) | **84.09** ($\pm$ 0.95) |
| | Dsub | 68.08 ($\pm$ 1.65) | 68.79 ($\pm$ 1.37) | 67.55 ($\pm$ 3.05) | **69.76** ($\pm$ 1.54) |
| DyGFormer | Wikipedia | 69.42 ($\pm$ 7.60) | 67.15 ($\pm$ 4.81) | 68.39 ($\pm$ 8.78) | **74.85** ($\pm$ 3.07) |
| | Dsub | 72.30 ($\pm$ 2.02) | 70.86 ($\pm$ 2.63) | 69.21 ($\pm$ 2.08) | **72.39** ($\pm$ 1.91) |

quantified as follows:

$$\hat{N}_{u'} = \max \left\{ k \in \mathbb{N}^+ \mid y_{u'}^{t_i} = y_{u'}^{T_{u'}}, \forall i \in \{|\mathcal{T}_{u'}| - k, \ldots, |\mathcal{T}_{u'}| - 1\} \right\}, \tag{11}$$

$$C_{u'} = \frac{\hat{N}_{u'}}{|\mathcal{T}_{u'}| - 1}. \tag{12}$$

where $u' \in \mathcal{V}_{\text{neg}}$, $\mathcal{V}_{\text{neg}} = \{u' | u' \in \mathcal{V}, y_{u'}^{T_{u'}} = 1\}$.

Our analysis reveals that the low performance of the DLS baseline stems from the extreme inconsistency of the original dynamic labels. In Wikipedia, dynamic labels for negative samples change abruptly, leading to a consistency measure of $C_{u'} \equiv 0$ for many samples. This sudden shift represents potentially noisy or instantaneous state changes that are detrimental to the model's temporal learning process. Conversely, simple pseudo-labeling like CFT enforces overly rigid continuity ($C_{u'} \equiv 1$), which causes feature misalignment by treating all historical steps as the final state.

In contrast, PTCL generates pseudo-labels with optimized and varying temporal consistency (Figure 4a, typically $0 < C_{u'} < 1$). This targeted intermediate consistency is crucial: it facilitates smooth and meaningful temporal transitions in node representations and creates a stabilized supervision signal that is qualitatively superior to the original, noisy dynamic labels. The necessity of our variational EM strategy, therefore, lies not in mere label recovery, but in the generation of a cleaner, temporally adaptive supervision signal for robust dynamic learning.

## 4.4 RQ3: TEMPORAL CURRICULUM LEARNING ANALYSIS

To comprehensively evaluate our Temporal Curriculum Learning design, we conduct a comparison experiment against the naive solution which uses all the generated pseudo-labels, and two commonly used baseline strategies to choose more reliable pseudo-labels in Curriculum Learning, as introduced in Section G.1:

- *Confidence Score Threshold (CST)* (Sun et al., 2019; He et al., 2022; Cascante-Bonilla et al., 2020): This method filters pseudo-labels based on their confidence scores, improving the overall quality of the labels.
- *Entropy of Softmax Trajectory (EST)* (Song et al., 2019; Pei et al., 2024): This method filters pseudo-labels using the entropy of the softmax trajectory, which is an accumulated distribution that summarizes the model's disagreement across training rounds.

As shown in Table 3, our Temporal Curriculum Learning consistently achieves the best AUC across all backbones and datasets, confirming its effectiveness. The Naive strategy underperforms PTCL by an average of 1.74 %, reflecting the noise in unfiltered pseudo-labels. CST and EST perform even worse, as their static filters fail to capture the temporal reliability of pseudo-labels. These results highlight the advantage of leveraging temporal dynamics for curriculum learning.

## 4.5 RQ4: CONVERGENCE AND EFFICIENCY ANALYSIS

We assess the convergence and efficiency of `PTCL` on Wikipedia using 5 backbones, with CFT as the baseline. As shown in Figure 4b, `PTCL` converges rapidly across all models: TGAT surpasses its baseline at the first iteration, others within 2–4. All reach peak AUC in 6–10 iterations, with DyGFormer showing the largest gain (+16.1%). Each EM iteration adds only $0.8\times$–$1.2\times$ training time, demonstrating both fast convergence and practical overhead.

## 5 RELATED WORK

### 5.1 DYNAMIC NODE CLASSIFICATION

Dynamic graph learning spans discrete-time (snapshot-based) and continuous-time paradigms. A large body of work develops powerful dynamic graph encoders that process temporal interactions for downstream prediction tasks. Representative continuous-time models include Temporal Graph Networks (TGN) (Rossi et al., 2020), which maintain learnable memory states to aggregate historical information, and TGAT (Xu et al., 2020), which introduces functional time encodings and temporal self-attention. Other approaches, such as DyRep (Trivedi et al., 2019), DBGNN (Qarkaxhija et al., 2022), and TGBase (Poursafaei et al.), further explore temporal point processes, higher-order temporal message passing, or feature-based temporal representations to model evolving node states.

Beyond representation learning, some works perform dynamic node classification under various supervision settings. Early models such as JODIE (Kumar et al., 2019) and DynPPE (Guo et al., 2021) use temporal embeddings to classify user or node states. More recent methods—e.g., TADGNN (Sun et al., 2022) and OTGNet (Feng et al., 2023)—explicitly propagate information over time-augmented structures or disentangle temporal class-dependent signals. HYDG (Ma et al., 2024) further incorporates hypergraph structures to capture high-order temporal dependencies among evolving node–hyperedge relations. DyGPrompt (Yu et al., 2025) introduces a dual-prompt framework that aligns temporal pre-training objectives with downstream dynamic classification tasks via evolving node–time patterns. PRES (Su et al., 2024) further improves the scalability of memory-based dynamic GNNs via a predict-to-smooth framework that supports substantially larger temporal batches. Notably, several studies do not require full supervision at all timestamps. For example, SAD (Tian et al., 2023)—a semi-supervised anomaly detection framework on dynamic graphs—leverages temporal GNNs with pseudo-label contrastive learning to exploit limited labeled nodes, demonstrating that temporal supervision can be significantly relaxed.

The above literature provides strong evidence that dynamic node prediction has been studied across representation learning, timestamp-wise classification, and semi-supervised settings. However, most existing approaches either assume labels distributed across multiple timestamps or rely on abundant anomaly labels. In contrast, our work focuses on the under-explored setting where only final-timestamp labels are available, a setting that naturally arises in real-world applications with high annotation costs and label uncertainty. Building on continuous-time graph encoders, we introduce a curriculum-guided pseudo-labeling mechanism tailored to label-scarce dynamic classification. Additional related work is provided in Appendix G.

## 6 CONCLUSION

In this work, we address dynamic node classification under limited labels by proposing `PTCL`, an extensible method based on temporally-weighted pseudo-labels and a variational EM framework. `PTCL` achieves up to 11.23% AUC/ACC gain across diverse real-world scenarios, validating the effectiveness of modeling temporal dynamics and our proposed Temporal Curriculum Learning. We also introduce the CoOAG dataset and FLiD framework to support practical evaluation. Beyond classification, `PTCL` is adaptable to other dynamic graph tasks, offering a solid foundation for future research on learning node evolution under realistic supervision constraints.

ETHICS STATEMENT

All authors of this paper have read and agree to adhere to the ICLR Code of Ethics. Our work involves the construction and release of a new academic collaboration dataset (CoOAG), derived from publicly available data in the Open Academic Graph (OAG). All author labels in CoOAG are inferred using a large language model (Qwen-Plus (Yang et al., 2024)) based on anonymized publication metadata (e.g., Fields of Study and abstracts), without accessing personally identifiable information. The labeling process was validated on a manually annotated subset (120 samples) and does not involve human subjects in experimental settings, thus no Institutional Review Board (IRB) approval was required. We do not foresee harmful applications of our method; on the contrary, `PTCL` is designed to reduce annotation costs in dynamic graph settings such as fraud detection or academic trend analysis, potentially benefiting resource-constrained institutions. The proposed framework FLiD is open-sourced to promote transparency and equitable access. No conflicts of interest exist among the authors, and this research received no external sponsorship.

REPRODUCIBILITY STATEMENT

We have taken extensive measures to ensure the reproducibility of our results. The complete implementation of our method `PTCL`, the FLiD framework, data preprocessing pipelines, and all experimental configurations are included in the supplementary materials. Hyperparameters for each backbone, training protocols, optimizer settings, and early stopping criteria are detailed in Appendix K (Table 10), and model architectures are described in Appendix E. The CoOAG dataset construction procedure, including prompt templates, feature extraction, and temporal splitting, is fully documented in Appendix D.2.2. All experiments were repeated over five random seeds, with mean and standard deviation reported in Table 1. Code and data will be made publicly available upon publication.

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

CONTENTS

## A  NOTATION TABLE

In this section, we summarize the notations used throughout the paper in Table 4.

| Symbol | Description |
|---|---|
| $\mathcal{G}$ | Dynamic graph consisting of temporally ordered events |
| $x(t_i) = (u_i, v_i, t_i)$ | Interaction event at timestamp $t_i$ |
| $\mathcal{V}$ | Set of nodes in the dynamic graph |
| $\mathcal{Y}$ | Set of possible class labels |
| $y_u^t$ | Label of node $u$ at timestamp $t$ |
| $\mathcal{T}_u$ | Set of timestamps at which node $u$ appears |
| $T_u = \max \mathcal{T}_u$ | Final timestamp of node $u$ |
| $\mathcal{Y}_F$ | Set of final timestamp labels for all nodes |
| $\mathcal{Y}_E$ | Set of labels at non-final timestamps |
| $T_B$ | Boundary timestamp separating train/eval partitions |
| $\mathcal{Y}_{F,B}$ | Final labels whose timestamps are before $T_B$ |
| $\mathcal{Y}_{F,A}$ | Final labels whose timestamps are after $T_B$ |
| $\mathcal{Y}_{E,B}$ | Non-final labels before $T_B$ (unknown in our setting) |
| $n_u \in \mathbb{R}^{d_N}$ | Node feature vector |
| $e_{u,v}^t \in \mathbb{R}^{d_E}$ | Edge feature at timestamp $t$ |
| $h_t^u$ | Backbone-generated embedding of node $u$ at timestamp $t$ |
| $\theta$ | Parameters of the dynamic graph backbone |
| $\phi$ | Parameters of the decoder used for label prediction |
| $\hat{\mathcal{Y}}_{E,B}$ | Pseudo-labels generated by the decoder |
| $\alpha, \beta$ | Hyperparameters controlling label weighting |
| $\tau$ | EM iteration index |
| $d_u^t$ | Temporal distance from timestamp $t$ to $T_u$ |
| $w_u^{t,\tau}$ | Temporal curriculum weight for node $u$ at timestamp $t$ in the $\tau$-th EM iteration |
| $\gamma$ | Decay coefficient for temporal curriculum learning |

Table 4: Summary of notations used in the paper.

## B  DERIVATIONS

### B.1  DERIVATION OF EQ. (1)

The objective of our Variational EM framework is to maximize the marginal log-likelihood of the observed data, $\log p_\theta(\mathcal{Y}_{F,B}|\mathcal{G})$. Since $\mathcal{Y}_{E,B}$ are the latent variables, we can express the marginal log-likelihood by introducing an arbitrary variational distribution $q_\phi(\mathcal{Y}_{E,B}|\mathcal{G})$:

$$
\begin{aligned}
\log p_\theta(\mathcal{Y}_{F,B}|\mathcal{G}) =& \mathbb{E}_{q_\phi(\mathcal{Y}_{E,B}|\mathcal{G})} \left[ \log p_\theta(\mathcal{Y}_{F,B}|\mathcal{G}) \right] \\
=& \mathbb{E}_{q_\phi(\mathcal{Y}_{E,B}|\mathcal{G})} \left[ \log \left( \frac{p_\theta(\mathcal{Y}_{F,B}, \mathcal{Y}_{E,B}|\mathcal{G})}{p_\theta(\mathcal{Y}_{E,B}|\mathcal{Y}_{F,B}, \mathcal{G})} \right) \right] \\
=& \mathbb{E}_{q_\phi(\mathcal{Y}_{E,B}|\mathcal{G})} \left[ \log \left( \frac{p_\theta(\mathcal{Y}_{F,B}, \mathcal{Y}_{E,B}|\mathcal{G})}{q_\phi(\mathcal{Y}_{E,B}|\mathcal{G})} \cdot \frac{q_\phi(\mathcal{Y}_{E,B}|\mathcal{G})}{p_\theta(\mathcal{Y}_{E,B}|\mathcal{Y}_{F,B}, \mathcal{G})} \right) \right] \\
=& \mathbb{E}_{q_\phi(\mathcal{Y}_{E,B}|\mathcal{G})} \left[ \log \left( \frac{p_\theta(\mathcal{Y}_{F,B}, \mathcal{Y}_{E,B}|\mathcal{G})}{q_\phi(\mathcal{Y}_{E,B}|\mathcal{G})} \right) \right] + \mathbb{E}_{q_\phi(\mathcal{Y}_{E,B}|\mathcal{G})} \left[ \log \left( \frac{q_\phi(\mathcal{Y}_{E,B}|\mathcal{G})}{p_\theta(\mathcal{Y}_{E,B}|\mathcal{Y}_{F,B}, \mathcal{G})} \right) \right] \\
=& \underbrace{\mathbb{E}_{q_\phi(\mathcal{Y}_{E,B}|\mathcal{G})} \left[ \log p_\theta(\mathcal{Y}_{F,B}, \mathcal{Y}_{E,B}|\mathcal{G}) - \log q_\phi(\mathcal{Y}_{E,B}|\mathcal{G}) \right]}_{\text{ELBO, } \mathcal{L}(\theta,\phi)} \\
& + \underbrace{D_{KL} \left( q_\phi(\mathcal{Y}_{E,B}|\mathcal{G}) \middle\| p_\theta(\mathcal{Y}_{E,B}|\mathcal{Y}_{F,B}, \mathcal{G}) \right)}_{\geq 0} \\
\geq& \mathbb{E}_{q_\phi(\mathcal{Y}_{E,B}|\mathcal{G})} \left[ \log p_\theta(\mathcal{Y}_{F,B}, \mathcal{Y}_{E,B}|\mathcal{G}) - \log q_\phi(\mathcal{Y}_{E,B}|\mathcal{G}) \right]
\end{aligned}
$$

Since the KL divergence $D_{KL}(q_\phi \| p_\theta) \geq 0$, the ELBO $\mathcal{L}(\theta, \phi)$ is indeed a lower bound on the marginal log-likelihood. The Variational EM algorithm iteratively maximizes this lower bound.

## B.2 DERIVATION OF EQ. (3)

The goal of the E-step in the Variational EM framework is to find the optimal variational distribution $q_\phi(\mathcal{Y}_{E,B}|\mathcal{G})$ that best approximates the true posterior distribution $p_\theta(\mathcal{Y}_{E,B}|\mathcal{Y}_{F,B}, \mathcal{G})$ by minimizing the Kullback-Leibler (KL) divergence $D_{KL}\left(q_\phi(\mathcal{Y}_{E,B}|\mathcal{G})\middle\| p_\theta(\mathcal{Y}_{E,B}|\mathcal{Y}_{F,B}, \mathcal{G})\right)$.

Directly optimizing this standard KL divergence is challenging because the entropy of the variational distribution $q_\phi(\mathcal{Y}_{E,B}|\mathcal{G})$ is generally computationally intractable to calculate.

Following prior work (Zhao et al., 2022; Qu et al., 2019), we adopt the wake-sleep algorithm (Hinton et al., 1995) for the E-step. This approach shifts the optimization focus to the reverse KL-divergence, $D_{KL}\left(p_\theta(\mathcal{Y}_{E,B}|\mathcal{Y}_{F,B}, \mathcal{G})\middle\| q_\phi(\mathcal{Y}_{E,B}|\mathcal{G})\right)$, which is equivalent to maximizing $\mathbb{E}_{p_\theta(\mathcal{Y}_{E,B}|\mathcal{Y}_{F,B}, \mathcal{G})}\left[\log q_\phi(\mathcal{Y}_{E,B}|\mathcal{G})\right]$:

$$
\begin{aligned}
&- D_{KL}\left(p_\theta(\mathcal{Y}_{E,B}|\mathcal{Y}_{F,B}, \mathcal{G})\middle\| q_\phi(\mathcal{Y}_{E,B}|\mathcal{G})\right) \\
&= \mathbb{E}_{p_\theta(\mathcal{Y}_{E,B}|\mathcal{Y}_{F,B}, \mathcal{G})}\left[\log q_\phi(\mathcal{Y}_{E,B}|\mathcal{G}) - \log p_\theta(\mathcal{Y}_{E,B}|\mathcal{Y}_{F,B}, \mathcal{G})\right] \\
&= \mathbb{E}_{p_\theta(\mathcal{Y}_{E,B}|\mathcal{Y}_{F,B}, \mathcal{G})}\left[\log q_\phi(\mathcal{Y}_{E,B}|\mathcal{G})\right] + \text{const},
\end{aligned}
$$

To furthur simplify the optimization, we apply the mean-field approximation to the variational distribution $q_\phi(\mathcal{Y}_{E,B}|\mathcal{G})$. Since the dynamic graph backbone $\theta$ is fixed during E-step, the node feature $\mathbf{h}_u^t$ is fixed too. Then we can get (Eq. (2) in the main paper):

$$
q_\phi(\mathcal{Y}_{E,B}|\mathcal{G}) = \prod_{u \in \mathcal{V}} \prod_{\substack{t \in \mathcal{T}_u \setminus \{T_u\} \\ t \leq T_B}} q_\phi(y_u^t|\mathbf{h}_u^t), \tag{13}
$$

Substituting this factorization into our objective gives the objective function $\hat{\mathcal{O}}_\phi$ (Eq. (3) in the main paper):

$$
\begin{aligned}
\hat{\mathcal{O}}_\phi &= \max_\phi \mathbb{E}_{p_\theta(\mathcal{Y}_{E,B}|\mathcal{Y}_{F,B}, \mathcal{G})}\left[\log q_\phi(\mathcal{Y}_{E,B}|\mathcal{G})\right] \\
&= \max_\phi \mathbb{E}_{p_\theta(\mathcal{Y}_{E,B}|\mathcal{Y}_{F,B}, \mathcal{G})}\left[\log\left(\prod_{u \in \mathcal{V}} \prod_{\substack{t \in \mathcal{T}_u \setminus \{T_u\} \\ t \leq T_B}} q_\phi(y_u^t|\mathbf{h}_u^t)\right)\right] \\
&= \max_\phi \mathbb{E}_{p_\theta(\mathcal{Y}_{E,B}|\mathcal{Y}_{F,B}, \mathcal{G})}\left[\sum_{u \in \mathcal{V}} \sum_{\substack{t \in \mathcal{T}_u \setminus \{T_u\} \\ t \leq T_B}} \log q_\phi(y_u^t|\mathbf{h}_u^t)\right] \\
&= \max_\phi \sum_{u \in \mathcal{V}} \sum_{\substack{t \in \mathcal{T}_u \setminus \{T_u\} \\ t \leq T_B}} \mathbb{E}_{p_\theta(y_u^t|\mathcal{Y}_{F,B}, \mathcal{G})}\left[\log q_\phi(y_u^t|\mathbf{h}_u^t)\right],
\end{aligned}
$$

## B.3 DERIVATION OF EQ. (7)

The M-step of the Variational EM framework aims to update the dynamic graph backbone parameters $\theta$ by maximizing the expected log-likelihood while holding the decoder parameters $\phi$ fixed, that is:

$$
\hat{\mathcal{O}}_\theta = \mathbb{E}_{q_\phi(\mathcal{Y}_{E,B}|G)}\left[\log p_\theta(\mathcal{Y}_{F,B}, \mathcal{Y}_{E,B}|\mathcal{G})\right],
$$

However, this expectation over the latent variables $\mathcal{Y}_{E,B}$ is analytically intractable. To make optimization feasible, we introduce two approximations.

**Pseudo-likelihood approximation.** Following (Besag, 1975), we approximate the joint by a product of conditional likelihoods:

$$\hat{\mathcal{O}}_\theta = \mathbb{E}_{q_\phi(\mathcal{Y}_{E,B}|\mathcal{G})} \left[\log p_\theta(\mathcal{Y}_{F,B}, \mathcal{Y}_{E,B} \mid \mathcal{G})\right]$$

$$\approx \mathbb{E}_{q_\phi(\mathcal{Y}_{E,B}|\mathcal{G})} \left[ \sum_{u\in\mathcal{V}} \sum_{t\in\mathcal{T}_u} \log p_\theta\left(y_u^t \mid \mathcal{G},\ \mathcal{Y}_{F,B} \cup \mathcal{Y}_{E,B} \setminus \{y_u^t\}\right)\right],$$

The pseudo-likelihood approximation replaces an intractable joint with a sum of conditionals, which makes per-label losses available for gradient-based optimization. Then we split the sum into (i) intermediate timestamps $t \in T_u \setminus \{T_u\}$ (pseudo-labels) and (ii) final timestamps $t = T_u$ (observed final labels). To control the relative contribution of pseudo-labels and final labels, we introduce a balancing hyperparameter $\beta \in [0, 1]$. Thus we approximate the expected log-likelihood by:

$$\hat{\mathcal{O}}_\theta \approx \mathbb{E}_{q_\phi(\mathcal{Y}_{E,B}|\mathcal{G})} \Big[\beta \sum_{u\in\mathcal{V}} \sum_{\substack{t\in\mathcal{T}_u\setminus\{T_u\}\\ t\le T_B}} \log p_\theta\left(y_u^t \mid \mathcal{G}, \mathcal{Y}_{F,B}, \mathcal{Y}_{E,B} \setminus \{y_u^t\}\right)$$

$$+ (1-\beta) \sum_{\substack{u\in\mathcal{V}\\ T_u\le T_B}} \log p_\theta\left(y_u^{T_u} \mid \mathcal{G}, \mathcal{Y}_{F,B} \setminus \{y_u^{T_u}\}, \mathcal{Y}_{E,B}\right)\Big].$$

Here we have explicitly conditioned each factor on the available final labels $\mathcal{Y}_{F,B}$ and on the unknown intermediate labels $\mathcal{Y}_{E,B}$.

**Monte Carlo approximation.** The expectation over $q_\phi(\mathcal{Y}_{E,B}|\mathcal{G})$ is still expensive. Using a Monte Carlo approximation with a single sample, which is common in practice, and denoting the sampled pseudo-label set by $\hat{\mathcal{Y}}_{E,B}$ as in the main text, we can get:

$$\hat{\mathcal{O}}_\theta = \beta \sum_{u\in\mathcal{V}} \sum_{\substack{t\in\mathcal{T}_u\setminus\{T_u\}\\ t\le T_B}} \log p_\theta(y_u^t|\mathcal{G}, \mathcal{Y}_{F,B}, \hat{\mathcal{Y}}_{E,B} \setminus \{\hat{y}_u^t\})$$

$$+ (1-\beta) \sum_{\substack{u\in\mathcal{V}\\ T_u\le T_B}} \log p_\theta(y_u^{T_u}|\mathcal{G}, \mathcal{Y}_{F,B} \setminus \{y_u^{T_u}\}, \hat{\mathcal{Y}}_{E,B}), \tag{14}$$

which is exactly Eq. (7) in the main paper. The Monte Carlo single-sample step removes the expectation over $q_\phi(\mathcal{Y}_{E,B} \mid \mathcal{G})$ and yields a deterministic training objective in each EM iteration, while a single sample is an efficient and effective choice. It can be seen as a stochastic unbiased estimator of the expectation.

## C  ALGORITHM FOR TRAINING PTCL

First, we define the link prediction task. Given a dynamic graph $\mathcal{G} = \{(u_i, v_i, t_i)\}$, the link prediction loss is defined as:

$$\mathcal{L}_{lp} = - \sum_{(u_i,v_i,t_i)\in\mathcal{G}} \log \sigma(\mathrm{MLP}(\mathbf{h}_{u_i}^{t_i} \parallel \mathbf{h}_{v_i}^{t_i})) - \sum_{(u_j,v_j,t_j)\notin\mathcal{G}} \log \left(1 - \sigma(\mathrm{MLP}(\mathbf{h}_{u_j}^{t_j} \parallel \mathbf{h}_{v_j}^{t_j}))\right). \tag{15}$$

where $\sigma(\cdot)$ is the sigmoid function, $\parallel$ means concatenation. The first term encourages the model to predict existing edges correctly, while the second term penalizes the model for predicting non-existent edges.

Then the algorithm for training PTCL can be summarized in Alg. 1.

## D  DATASET DETAILS

Due to the lack of widely studied datasets for the **label-limited dynamic node classification**, we utilize three existing datasets that align closely with our task and propose a new dataset *CoOAG* specifically designed for this problem. The statistics of four datasets are introduced in Table 5.

---

**Algorithm 1** Optimization Algorithm

---

1: **Input:** A dynamic graph $\mathcal{G}$, final timestamp labels $\mathcal{Y}_{F,B}$, and hyperparameter $\beta$
2: **Output:** Predicted $\hat{\mathcal{Y}}_{F,A}$.
3: $\theta \leftarrow \arg\min_\theta \mathcal{L}_{lp}$          ▷ Warm up the backbone
4: $\tau \leftarrow 1$          ▷ Initialize iteration counter
5: **repeat**
6:      **E-Step: Decoder Optimization**
7:          $\phi \leftarrow \arg\max_\phi \mathcal{O}_\phi$          ▷ Update the decoder $q_\phi$
8:          $\hat{\mathcal{Y}}_{E,B} \leftarrow \arg\max q_\phi(\mathcal{Y}_{E,B}|\mathcal{G})$          ▷ Generate pseudo-labels
9:      **M-Step: Backbone Optimization**
10:          $w_u^{t,\tau} \leftarrow f_{\text{TW}}(t, u, \tau, T_u, \gamma)$          ▷ Compute temporal weights
11:          $\theta \leftarrow \arg\max_\theta \mathcal{O}_\theta$          ▷ Update backbone $p_\theta$
12:      $\tau \leftarrow \tau + 1$          ▷ Update iteration counter
13: **until** Converged
14: $\hat{\mathcal{Y}}_{F,A} \leftarrow \arg\max q_\phi(\mathcal{Y}_{F,A}|\mathcal{G})$          ▷ Final prediction
15: **return** $\hat{\mathcal{Y}}_{F,A}$

---

Table 5: Dataset Statistics.

|  | Wikipedia | Reddit | Dsub | CoOAG |
|---|---|---|---|---|
| Nodes | $9,227$ | $10,984$ | $150,000$ | $9,559$ |
| Edges | $15,7474$ | $67,2447$ | $16,8154$ | $11,4337$ |
| Duration | 1 month | 1 month | 1 year | 22 years |
| Total classes | 2 | 2 | 2 | 5 |
| Bipartite | ✓ | ✓ | ✗ | ✗ |
| Node Feat Dim | – | – | 34 | 384 |
| Edge Feat Dim | 172 | 172 | 1 | 384 |

### D.1 PREVIOUS DATASETS

We use three publicly available datasets and do the preprocessing to adapt them to our task:

#### D.1.1 DSUB

**Description**: Dsub is a subgraph of the Dgraph (Huang et al., 2022) dataset, which is a financial fraud detection dataset where nodes represent users, and edges represent emergency contact relationships between users. Node labels indicate whether a user is ultimately identified as fraudulent (failing to repay loans over an extended period). Node features are derived from user metadata. In addition to confirmed fraudulent and non-fraudulent labels, the dataset includes background nodes that lack sufficient information for labeling but are retained to maintain graph connectivity.

**Preprocessing**: To facilitate efficient training, we extract a subgraph called Dsub using Breadth-First Search (BFS), ensuring that the subgraph remains connected and preserves the original label distribution.

#### D.1.2 WIKIPEDIA

**Description**: Wikipedia (Kumar et al., 2019) is a bipartite interaction graph that records edits on Wikipedia pages over one month. Nodes represent users and pages, and edges denote editing behaviors with timestamps. Each edge is associated with a 172-dimensional Linguistic Inquiry and Word Count (LIWC) feature. The dataset includes dynamic labels indicating whether users are temporarily banned from editing (Yu et al., 2023).

**Task Adaptation**: To simulate real-world scenarios where only the final labels are available, we split the dynamic labels into $\mathcal{Y}_{F,B}$ (final labels) and $\mathcal{Y}_{E,B}$ (unobserved labels). During training, only $\mathcal{Y}_{F,B}$ is used.

### D.1.3 REDDIT

**Description**: Reddit (Kumar et al., 2019) is a bipartite graph that records user posts under subreddits over one month. Nodes represent users and subreddits, and edges represent timestamped posting requests. Each edge is associated with a 172-dimensional LIWC feature. The dataset includes dynamic labels indicating whether users are banned from posting (Yu et al., 2023).

**Task Adaptation**: Similar to Wikipedia, we split the dynamic labels into $\mathcal{Y}_{F,B}$ and $\mathcal{Y}_{E,B}$, using only $\mathcal{Y}_{F,B}$ for training to simulate real-world conditions.

## D.2 CoOAG

### D.2.1 DESCRIPTION

To advance research in this domain, we introduce CoOAG, a novel dataset derived from the academic sphere, inspired by the Coauthor CS and Coauthor Physics networks (Shchur et al.). This dataset has undergone stringent quality control and temporal consistency checks. Label distributions are detailed in Table 6.

The CoOAG dataset is constructed using the Microsoft Academic Graph (MAG) portion from Open Academic Graph 2.1(Sinha et al.; Zhang et al., b;a; Tang et al.), with a focus on publications from leading AI conferences. The node labels in CoOAG denote authors' research interests, classified into the following categories:

- CV (Computer Vision)
- NLP (Natural Language Processing)
- ROB (Robotics)
- DM/WS (Data Mining/Web Search)
- AI/ML (Other AI Fields)

Table 6: Label Distributions of CoOAG.

| Field | Label Distribution |
|-------|--------------------|
| ROB | 2,845 (29.64%) |
| CV | 1,700 (17.71%) |
| NLP | 1,652 (17.21%) |
| AI/ML | 1,971 (20.53%) |
| DM/WS | 1,431 (14.91%) |

### D.2.2 PREPROCESSING

Although the Open Academic Graph (OAG) provides research interest tags, these annotations are static and fail to capture the temporal evolution of scholars' research trajectories. To address this limitation, we leverage structured prompts with the Qwen-Plus API (qwe, 2024; Yang et al., 2024) to dynamically classify authors' research interests based on their most recent ten publications, incorporating both abstracts and their associated Fields of Study (FoS). The prompt template, as illustrated in Listing D.2.2, encompasses:

- Category definitions with canonical examples
- Strict output format constraints
- Weighted keyword matching logic
- Interactive classification examples

This approach achieves 98.3% ACC on 120 manually verified samples. Edge features are generated by concatenating paper metadata and abstracts, encoded using the all-MiniLM-L12-v2 model. Node features are computed as the average of all paper features for each author. Conference submission deadlines determine edge timestamps. The classification workflow maintains temporal consistency by processing papers in chronological order.

```
Research field Classification Prompt Template
───────────────────────────────────────────────────────
Classify the author's research field into one of the following

5 categories based on the given field keywords and weights:
- 0: CV (Computer Vision)
- 1: NLP (Natural Language Processing)
- 2: ROB (Robotics)
- 3: DM/WS (Data Mining/Web Search)
- 4: AI/ML (Other AI Fields)

Input: Multiple field keywords with weights
Output requirements:
    - **(*@\textbf{Format}@*)**: Directly return classification

    result (0-4)
    - **(*@\textbf{Constraint}@*)**: Answer must be a single digit

    without explanation

Examples:
    - Input: "[computer vision (0.53377)] [image filter (0.5337)]"
    - Output: 0

Input:
    {fos_text}
───────────────────────────────────────────────────────
```

### D.2.3 EXAMPLES

To illustrate the temporal dynamics inherent in the CoOAG dataset, we present a concrete example of label evolution for a single author node. Consider **Node ID: 6816**, a researcher whose publication history spans 3,693 days (approximately 10 years). This node undergoes three label transitions across distinct research fields, reflecting meaningful shifts in academic focus:

Table 7: Label transitions of node id: 6816 over time.

| Time interval | Number of papers | Research field (label) |
| --- | --- | --- |
| 2010.11.15 – 2012.09.10 | 5 | Robotics (ROB) |
| 2012.09.10 – 2014.12.15 | 7 | Data Mining / Web Search (DM/WS) |
| 2014.12.15 – 2016.12.04 | 6 | Robotics (ROB) |
| 2016.12.04 – 2020.12.25 | 24 | Data Mining / Web Search (DM/WS) |

This trajectory demonstrates repeated and substantial shifts between Robotics and Data Mining/Web Search, underscoring the non-stationary nature of research interests over time. Such patterns are not isolated: across the entire CoOAG dataset, 43.2% of labeled nodes experience at least one label transition during their publication lifetime, with an average of 1.19 label changes per node. These statistics confirm that label dynamics in CoOAG are both frequent and semantically meaningful, capturing real-world academic evolution.

## E BACKBONE DETAILS

- **TGAT** (Xu et al., 2020) leverages a self-attention mechanism to simultaneously capture spatial and temporal dependencies. Initially, TGAT combines the raw node feature $\mathbf{n}_u$ with a learnable time encoding $z(t)$, forming $\mathbf{n}_u(t) = [\mathbf{n}_u || z(t)]$, where $z(t) = \cos(tw + b)$. Subsequently, self-attention is applied to generate the representation of node $u$ at time $t_0$, denoted as $\mathbf{h}_u^{t_0} = \mathrm{SAM}(\mathbf{n}_u(t_0), \{\mathbf{n}_v(m_v) \mid v \in N_{t_0}(u)\})$. Here, $N_{t_0}(u)$ represents the set of neighbors of node $u$

at time $t_0$, and $m_v$ indicates the timestamp of the most recent interaction involving node $v$. Finally, predictions for any node pair $(u, v)$ at time $t_0$ are obtained via MLP($[\mathbf{h}_u^{t_0} || \mathbf{h}_v^{t_0}]$).

- **TCL** (Wang et al., 2021a) adopts a contrastive learning framework. To construct interaction sequences for each node, TCL employs a breadth-first search algorithm on the temporal dependency subgraph. A graph transformer is then utilized to learn node representations by jointly considering graph topology and temporal dynamics. Additionally, TCL integrates a cross-attention mechanism to model the interdependencies between interacting nodes.

- **TGN** (Rossi et al., 2020) combines RNN-based and self-attention-based techniques. TGN maintains a memory module to store and update the state $s_u(t)$ of each node $u$, which serves as a compact representation of $u$'s historical interactions. Given the memory updater as mem, when an edge $e_{uv}(t)$ connecting nodes $u$ and $v$ is observed, the memory state of node $u$ is updated as $s_u(t) = \text{mem}(s_u(t^-), s_v(t^-) || \mathbf{e}_{u,v}^t)$, where $s_u(t^-)$ denotes the memory state of $u$ just prior to time $t$, and $\mathbf{e}_{u,v}^t$ represents the edge feature. The memory updater mem is implemented using a recurrent neural network (RNN). Node embeddings $\mathbf{h}_u^t$ are computed by aggregating information from the $L$-hop temporal neighborhood through self-attention.

- **GraphMixer** (Cong et al., 2023) introduces a simple yet effective MLP-based architecture. Instead of relying on trainable time encodings, GraphMixer utilizes a fixed time encoding function, which is integrated into a link encoder based on MLP-Mixer to process temporal links. A node encoder with neighbor mean-pooling is employed to aggregate node features. Specifically, for each node $u$, GraphMixer computes its embedding $\mathbf{h}_u^t$ by summarizing the features of its neighbors within the temporal context.

- **DyGFormer** (Yu et al., 2023) employs a self-attention mechanism to model dynamic graphs. For a given node $u$, DyGFormer retrieves the features of its involved neighbors and edges to represent their encodings. It incorporates a neighbor co-occurrence encoding scheme, which captures the frequency of each neighbor's appearance in the interaction sequences of both the source and destination nodes, thereby explicitly exploring pairwise correlations. Rather than operating at the interaction level, DyGFormer divides the interaction sequences of each source or destination node into multiple patches, which are then processed by a transformer to compute node embeddings $\mathbf{h}_u^t$.

## F MORE EXPERIMENTS

To further validate our framework, we additionally investigate the following research questions (RQs):

**RQ5**: How does decoder design impact `PTCL`'s performance? **RQ6**: Is `PTCL` robust to the choice of temporal decay rate $\gamma$?

### F.1 RQ5: DECODER COMPARISON

In our framework, we follow the modular paradigm commonly adopted in temporal graph learning, where the encoder (backbone) captures dynamic structural-temporal features, and the decoder serves as a lightweight task-specific mapping function from node embeddings to labels (Rossi et al., 2020). While pseudo-labels play a critical role in training under the label-limited setting, their quality is primarily determined by the encoder's representation capacity rather than the complexity of the decoder.

To further examine the role of the decoder, we conducted additional experiments with more expressive designs, including deep 8-layer MLP and Transformer-based architectures. Interestingly, these complex decoders often resulted in performance degradation, which can be attributed to overfitting, increased variance, or optimization instability under the low-label regime. Figure 5 summarizes the performance comparison on the Wikipedia dataset, showing that our lightweight MLP decoder achieves competitive or superior results across backbones compared with deep MLP and Transformer decoders.

### F.2 RQ6: HYPERPARAMETERS SENSITIVITY

To evaluate the sensitivity of our framework to the temporal decay rate $\gamma$, we conducted additional experiments on the Wikipedia dataset by varying $\gamma$ from 0.1 to 0.9. We report results across five different backbones in Table 8. Overall, the performance remains relatively stable across settings, with moderate standard deviations: TGAT ($83.15 \pm 1.39$), TGN ($86.25 \pm 0.76$), GraphMixer ($82.47$

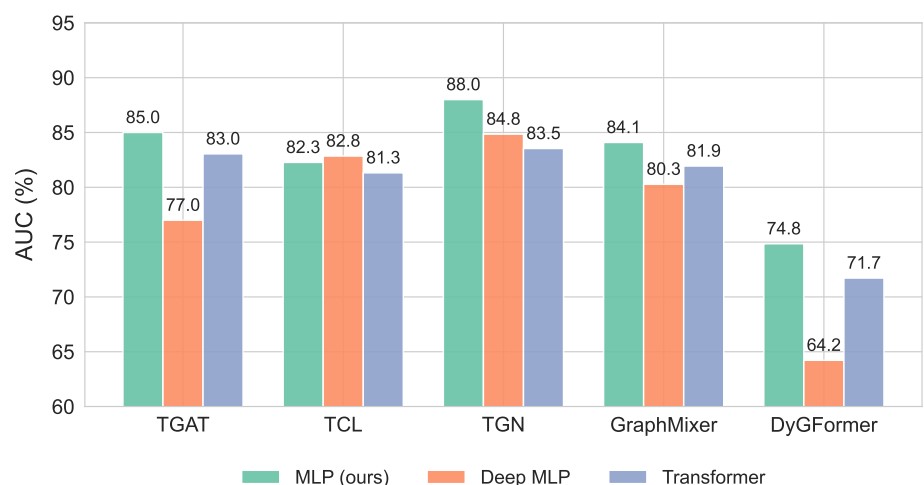

Figure 5: Decoder Comparison on Wikipedia.

$\pm\ 0.57$), TCL ($80.30 \pm 1.75$), and DyGFormer ($71.85 \pm 2.24$). These results suggest that our method is robust to the choice of $\gamma$.

Table 8: Sensitivity analysis of our framework to the temporal decay rate $\gamma$ on the Wikipedia dataset.

| Model | 0.1 | 0.2 | 0.3 | 0.4 | 0.5 | 0.6 | 0.7 | 0.8 | 0.9 | Avg $\pm$ Std |
|---|---|---|---|---|---|---|---|---|---|---|
| TGAT | 80.94 | 81.58 | 84.24 | 83.13 | 84.48 | 83.15 | 81.94 | 85.52 | 83.40 | $83.15 \pm 1.39$ |
| TGN | 85.58 | 87.03 | 85.27 | 87.01 | 86.57 | 87.17 | 85.03 | 86.09 | 86.52 | $86.25 \pm 0.76$ |
| GraphMixer | 82.37 | 81.23 | 83.44 | 82.27 | 82.92 | 82.36 | 82.86 | 82.50 | 82.30 | $82.47 \pm 0.57$ |
| TCL | 76.69 | 80.41 | 81.97 | 81.31 | 80.93 | 82.27 | 81.26 | 79.90 | 78.00 | $80.30 \pm 1.75$ |
| DyGFormer | 73.64 | 73.74 | 72.79 | 68.73 | 73.90 | 74.49 | 70.29 | 70.84 | 68.24 | $71.85 \pm 2.24$ |

## G MORE RELATED WORK

### G.1 PSEUDO-LABELING

Pseudo-labeling (Lee et al., 2013) is a widely used semi-supervised learning method that assigns labels to unlabeled data to reduce entropy and encourage low-density decision boundaries (Pei et al., 2024; Chapelle & Zien, 2005; Grandvalet & Bengio, 2004). It has proven effective in various fields including computer vision (Lee et al., 2013; Xu et al., 2021), graph learning (Li et al., 2022b; 2018b;a; Sun et al., 2019), knowledge distillation (Hinton et al., 2015), and adversarial training (Miyato et al., 2017; Xie et al., 2019). However, performance heavily depends on label quality, as noisy pseudo-labels may misguide training (Pei et al., 2024). Curriculum Learning (Bengio et al., 2009) mitigates this by ordering training samples from easy to hard, using heuristics like confidence or entropy (Cascante-Bonilla et al., 2020; He et al., 2022; Pei et al., 2024; Song et al., 2019). In contrast, our **Temporal Curriculum Learning** explicitly leverages temporal information to prioritize recent (easier) timestamps first, then gradually incorporates earlier (harder) ones, better aligning model training with evolving dynamics in time-dependent graphs.

A closely related work is ELI (Kamhoua et al.), which also addresses graph learning under limited supervision by introducing pseudo-labels and decoupling their generation from final label supervision. Despite this similarity, there are key differences. First, ELI is designed for static graphs with globally sparse labels, whereas our work considers dynamic graphs where labels are typically only available at the final timestamp, introducing temporally-driven label scarcity. Second, ELI infers pseudo-labels through an unsupervised label distribution estimation and constructs a pseudo-label graph, while our `PTCL` framework treats pseudo-labels as latent variables and refines them iteratively via a variational

EM procedure, with a decoder trained solely on ground-truth labels to ensure alignment with the true label space. Finally, ELI integrates labels through graph-based regularization using the Laplacian of the pseudo-label graph, whereas `PTCL` directly supervises the backbone with both pseudo- and ground-truth labels, augmented by a temporal curriculum weighting scheme that prioritizes recent timestamps. These distinctions highlight the novelty of our temporal perspective in pseudo-label integration for dynamic graphs.

### G.2 VARIATIONAL EM FRAMEWORK

The variational EM framework (Dempster et al., 1977; Neal & Hinton, 1998) is a widely used framework for parameter estimation in probabilistic models with latent variables. In the classical EM algorithm, the goal is to maximize the likelihood of observed data by iteratively refining model parameters through alternating E-steps (expectation computation) and M-steps (parameter maximization). GMNN (Qu et al., 2019) applies EM for semi-supervised static graphs and GLEM (Zhao et al., 2022) combines GNNs with language models. Our contribution lies not in framework innovation but in adapting this established paradigm to address label-limited dynamic node classification.

## H RELATIONSHIP WITH OTHER TASKS

In Section 2, we define the *label-limited dynamic node classification* task. Specifically, when all timestamps $t_i$ are identical or omitted, the graph degenerates into a static graph (Kipf & Welling, 2016; Holme & Saramäki, 2012), where each node $u \in \mathcal{V}$ is associated with a single label $y_u$. Then the problem degrades to a *static node classification* task. Alternatively, if labels are available for all nodes at all timestamps, i.e., $\mathcal{Y}_E$ is entirely known, the problem becomes a *fully supervised dynamic node classification* task, which has been extensively studied in prior research (Xu et al., 2020; Rossi et al., 2020; Cong et al., 2023; Yu et al., 2023).

## I DISCUSSION

**Limitation.** Due to the open-access nature and the unique characteristics of our task setting, we were unable to evaluate our method on a broader range of datasets. This may limit the generalizability of our findings to other domains or data types. Additionally, we observed that performance can be influenced by the specific computational environment, including hardware and runtime configurations. As a result, reproduction under different resource settings may lead to variations in absolute performance.

**Future Work.** Although our Temporal Curriculum Learning outperforms alternative dynamic weighting strategies based on confidence or entropy (e.g., CST and EST), we acknowledge that our current design still relies on a manually set temporal decay parameter $\gamma$. A valuable future direction is to develop a confidence-aware weighting scheme that adaptively modulates pseudo-label contributions according to their reliability. In addition, we plan to explore the integration of large language models (LLMs) into our framework, leveraging their strong reasoning and representation capabilities to further enhance pseudo-label generation and temporal modeling in dynamic graphs.

## J IMPLEMENTATION DETAILS

We use PyTorch (Paszke et al., 2019), scikit-learn (Pedregosa et al., 2011), PyTorch Geometric (Fey & Lenssen, 2019), DyGLib (Yu et al., 2023) library to implement our proposed framework FLiD. We conduct experiments on two clusters: (1) 4×Tesla V100 (32GB memory) using 16-core CPUs and 395GB RAM; (2) 8×2080Ti (11GB memory) using 12-core CPUs and 396GB RAM.

## K HYPERPARAMETERS

We optimize all methods across all models using the Adam optimizer (Kingma & Ba, 2014), with cross-entropy loss as the objective function. Initially, we warm up all backbones through link prediction tasks (Kumar et al., 2019). Subsequently, the entire models are trained for 100 epochs, employing an early stopping strategy with a patience of 15. For consistency, we set the learning

rate to 0.0001 and the batch size to 200 across all methods and datasets. To ensure robustness and minimize deviations, we conduct five independent runs for each method with random seeds ranging from 0 to 4 and report the average performance (Yu et al., 2023).

### K.1   MODEL CONFIGURATIONS

Here, we present the configurations for each model (Table 9): TGAT, TGN, TCL, GraphMixer, and DyGFormer, all of which remain consistent across datasets.

### K.2   PTCL HYPERPARAMETERS

Here, we present the hyperparameters of PTCL (Table 10): $\beta$ is a hyperparameter that balances the weight of pseudo-labels and final timestamp labels; $\gamma$ is a hyper-parameter that controls the rate of Temporal Curriculum decay. Note that TGN runs out of memory on Dsub due to its high space cost.

Table 9: Model Configurations Comparison.

| Hyperparameter | TGAT | TGN | TCL | GraphMixer | DyGFormer |
|---|---|---|---|---|---|
| Time encoding dim | 100 | 100 | 100 | 100 | 100 |
| Output dim | 172 | 172 | 172 | 172 | 172 |
| Attention heads | 2 | 2 | 2 | – | 2 |
| Graph conv layers | 2 | 1 | – | – | – |
| Transformer layers | – | – | 2 | – | 2 |
| MLP-Mixer layers | – | – | – | 2 | – |
| Node memory dim | – | 172 | – | – | – |
| Depth encoding dim | – | – | 172 | – | – |
| Co-occurrence dim | – | – | – | – | 50 |
| Aligned encoding dim | – | – | – | – | 50 |
| Memory updater | – | GRU | – | – | – |
| Time gap $T$ | – | – | – | 2000 | – |

Table 10: Hyperparameters of PTCL.

| Model | Hyperparameters | Wikipedia | Reddit | Dsub | CoOAG |
|---|---|---|---|---|---|
| TGAT | $\beta$ | 0.9 | 0.9 | 0.7 | 0.2 |
| | $\gamma$ | 0.8 | 0.1 | 0.2 | 0.9 |
| TCL | $\beta$ | 0.1 | 0.9 | 0.1 | 0.8 |
| | $\gamma$ | 0.6 | 0.9 | 0.5 | 0.6 |
| TGN | $\beta$ | 0.9 | 0.9 | – | 0.9 |
| | $\gamma$ | 0.05 | 0.01 | – | 0.1 |
| GraphMixer | $\beta$ | 0.5 | 0.5 | 0.5 | 0.5 |
| | $\gamma$ | 1.3 | 0.1 | 0.1 | 0.4 |
| DyGFormer | $\beta$ | 0.7 | 0.5 | 0.1 | 0.1 |
| | $\gamma$ | 0.01 | 0.01 | 0.5 | 0.1 |

## L   THE USE OF LARGE LANGUAGE MODELS (LLMs)

We employed large language models (LLMs) as auxiliary tools to enhance the clarity and readability of the text. Specifically, LLMs were used to assist in identifying and correcting grammar and spelling errors, as well as refining the overall expression of our writing. The role of LLMs was limited to language polishing, ensuring that the final manuscript meets academic writing standards.

