# OpenReview forum: "PTCL: Pseudo-Label Temporal Curriculum Learning for Label-Limited Dynamic Graph"
_ICLR.cc/2026/Conference — ICLR 2026 Conference Withdrawn Submission_

### Official Review · Reviewer_DohM · 2025-10-23

**Soundness:** 3
**Presentation:** 3
**Contribution:** 4
**Rating:** 6
**Confidence:** 4

**Summary:**

In this work, the authors focus on dynamic node classification task in temporal graph under the limited label setting. The paper proposed PTCL, an extensible method with temporally-weighted pseudo-labels and a variational EM framework. The authors also included the CoOAG dataset, a novel benchmark for this task.
Across four datasets, the authors showed that PTCL improves various base TGNN's performance. The unified FLiD framework is also a nice contribution for future work.

**Strengths:**

There has long been a lack of study for node level task in temporal graph learning. This work help bridge this gap by focusing on the dynamic node classification task. specifically it has the following strength:

- **significance: problem that needed attention**. As mentioned, the dynamic node classification problem has been under-studied in the literature and this work focus on this task which will be beneficial for the community.

- **extensive evaluation**. Generally there is a lack of node classification dataset in temporal graph due to lack of dynamic labels (as the authors mentioned). In this work, the authors evaluated on four datasets with a range of temporal graph methods across different baseline training approaches. The original dataset is also a nice contribution.

- **clear presentation**. The paper is clearly presented and easy to follow

**Weaknesses:**

- **computational complexity**: as datasets in this work are on the smaller side (less than a million edges). It would be good for the authors to show the complexity of the approach and the compute time. If there is larger labeled datasets, it would be good to benchmark on it as well.

- **Figure 3 is not clear**: figure 3 visualisation is not clear and end up being more confusing than just reading the description in Section 4.1.2. For example, what does the red box mean? Also B and D to represent backbone and decoder is confusing at a glance. Please update the figure or remove it.

- **lack of detail for FLiD in the main paper**. Potentially adding a diagram to make FLiD more clear and how it helps facilitate the process.

**Questions:**

- **Dsub** why take a subset of the **Dgraph** dataset?

**Details Of Ethics Concerns:**

No ethical review needed.

---

> ### Author Response · Authors · 2025-11-26
> **W1 Response regarding computational complexity and scalability**
>
> ## W1 Response regarding computational complexity and scalability
> We thank the reviewer for raising this important point. Our response consists of two parts: (1) computational complexity and runtime analysis of PTCL, and (2) experiments on larger-scale dynamic graphs.
>
> ### **1. Computational Complexity and Runtime**
>
> We appreciate the reviewer’s concern about the efficiency of PTCL. As noted, our paper already reports empirical training-time behavior in Section 4.5 (Figure 4b). Specifically, we benchmarked PTCL on Wikipedia across five backbone models. The results show that PTCL converges quickly—typically within **2–4 EM iterations** —with each iteration adding only a modest overhead of **0.8×–1.2×** relative to the base model’s training time. TGAT exceeds its baseline after a single iteration, and all remaining models reach peak accuracy within **6–10 iterations**, indicating that PTCL preserves practical training efficiency while yielding substantial performance gains.
>
> From a theoretical standpoint, the main additional cost comes from the EM loop. PTCL increases the per-epoch complexity from **O(n)** to **O(kn)**, where k is the number of EM iterations. Because k is a small constant (≪ n) determined empirically and convergence is fast, this increase corresponds to only a constant-factor overhead.
>
> Memory usage is also stable: PTCL stores pseudo-labels and their weights, whose sizes scale with the number of nodes and are reused across iterations. Thus, the memory footprint remains comparable to standard dynamic graph training.
>
> ### **2. Experiments on Larger Dynamic Graphs**
>
> Regarding the reviewer’s suggestion to evaluate on larger labeled datasets, we fully agree. To address this, we have sampled a new large-scale dynamic graph from the **DGraph** collection, containing **1.09 million edges**, which is significantly larger than the datasets in the current version. We are currently running PTCL on this dataset to benchmark runtime, scalability, and performance in a higher-volume setting.
>
> We will integrate the results into the updated experimental section as soon as the runs are complete.

---

> ### Author Response · Authors · 2025-11-26
> **W2 Modified Figure 3**
>
> ## W2 Modified Figure 3
> We are grateful for the valuable feedback from the reviewer that figure 3 visualisation is somehow confusing, and we have updated it in the revised version. To be specific, the red box means **'being jointly optimized'**, and we update it to the caption of the figure. Moreover, we use 'BB' and 'Dec' to represent backbone and decoder respectively. We believe these revisions make Figure 3 a much more effective tool for understanding our proposed methodology.

---

> ### Author Response · Authors · 2025-11-26
> **W3 Detailed Description of FLiD**
>
> ## W3 Detailed Description of FLiD
> Thank you for highlighting that the description of **FLiD** in the main paper lacked sufficient detail. We agree that a clearer presentation can help readers better understand the framework and its role in our setting.
>
> To address this, we have moved the detailed explanation of FLiD from the appendix into the main text. Specifically, the expanded content is now incorporated into Section 4.1.3, where we substantially enhance the original FLiD description.
>
> The updated section now clearly covers:
> - The motivation of FLiD relative to existing dynamic graph learning frameworks (DyGLib, TGL);
> - The design goals of FLiD for the label-limited dynamic node classification setting;
> - Its support for multiple training paradigms (CFT, DLS, NPL, SEM, PTCL, PTCL-2D);
> - Its pseudo-labeling enhancements (CST, EST, temporal curriculum learning);
> - Compatibility with several dynamic graph backbones;
> - The preprocessing and splitting strategy tailored for final-timestamp-only supervision.

---

> ### Author Response · Authors · 2025-11-26
> **W4 Explanation of Dsub Dataset**
>
> ## W4 Explanation of Dsub Dataset
> We appreciate the reviewer’s question regarding the use of a subset of the DGraph dataset. The original DGraph is extremely large and highly sparse, which makes it challenging for many backbone models to train efficiently or even converge properly under the label-limited setting. To ensure stable and efficient training across all baseline backbones, we extract a connected subgraph using BFS. This provides a denser and computationally manageable graph while preserving the temporal and structural properties needed for our setting.
>
> In addition, as noted in our response to Reviewer W1, we have also extracted a **much larger** BFS-based subgraph (approximately **1 million edges**) to further verify that our method generalizes well to larger graph scales. We are currently running experiments on this extended dataset and will update the results in the revised version of the paper as soon as they are available.

---

### Official Review · Reviewer_68ux · 2025-10-26

**Soundness:** 3
**Presentation:** 2
**Contribution:** 2
**Rating:** 4
**Confidence:** 4

**Summary:**

The paper identifies a key challenge of node classification in dynamic graph learning, where only the final-time labels are collected and the intermediate labels are missing. To address this challenge, the authors propose a framework that dynamically generates pseudo labels for intermediate time steps and progressively refines them during training. The proposed method is evaluated on several benchmark datasets, demonstrating its effectiveness compared to existing baselines.

**Strengths:**

1. The paper identifies a key challenge of node classification in dynamic graph learning, where only the final-time labels are collected and the intermediate labels are missing. Such a challenge is common in real-world applications but has been largely overlooked by existing works.
2. The paper proposes a framework that dynamically generates pseudo labels for intermediate time steps and progressively refines them during training. The proposed method is well-motivated.

**Weaknesses:**

1. The cited related works are not recent enough.
2. Although the paper made efforts in clarifying its theoretical foundation, it is still hard to follow.
3. Please avoid leaving so many dangling lines that have only one word in the last line of a paragraph.

**Questions:**

1. Why is using the true dynamic labels perform worse than the pesudo labels? The true labels should be the upper-bound performance.
2. Since ELI is nominated as a competing work, why there are no comparisons with it?
3. As shown in Table 1, the baseline that uses true dynamic labels (DLS) does not have significant improvements over the baselines that use pseudo labels (CFT and NPL). To be honest, it is worse in many cases. Then what is the necessity of recovering all those labels? It seems that the curriculum learning strategy is the key to the performance boost, then the contribution is largely reduced.

---

> ### Author Response · Authors · 2025-11-26
> **W1 More Related Work**
>
> ## W1 More Related Work
> We thank the reviewer for pointing this out. Following your suggestion, we have provided a substantially expanded and more comprehensive related work section on dynamic graphs and dynamic node classification, which now includes several important recent advances. In particular, beyond classical temporal embedding methods (e.g., JODIE, DynPPE) and more recent time-augmented or class-disentangled models (e.g., TADGNN [1], OTGNet), we now incorporate:
> 1. **HYDG** [2], which introduces hypergraph-based temporal modeling to capture high-order node–hyperedge dynamics;
> 2. **DyGPrompt** [3], which proposes a dual-prompt mechanism that aligns temporal pre-training with downstream dynamic node classification;
> 3. **PRES** [4], which improves the scalability of memory-based DGNNs through a predict-to-smooth training framework; and
> 4. **SAD** [5], a semi-supervised dynamic anomaly detection framework that leverages pseudo-label contrastive learning and demonstrates that full temporal supervision is not required.
> These additions allow us to more accurately situate our contribution within the dynamic node classification literature and address the reviewer’s concern regarding completeness.
>
> ### **References**
> [1] Jiarui Sun, Mengting Gu, Chin-Chia Michael Yeh, Yujie Fan, Girish Chowdhary, and Wei Zhang. Dynamic graph node classification via time augmentation, 2022.
>
> [2] Xiaoxu Ma, Chen Zhao, Minglai Shao, and Yujie Lin. Hypergraph-based dynamic graph node classification, 2024.
>
> [3] Xingtong Yu, Zhenghao Liu, Xinming Zhang, and Yuan Fang. Node-time conditional prompt learning in dynamic graphs. In The Thirteenth International Conference on Learning Representations, 2025.
>
> [4] Junwei Su, Difan Zou, and Chuan Wu. Pres: Toward scalable memory-based dynamic graph neural networks, 2024.
>
> [5] Sheng Tian, Jihai Dong, Jintang Li, Wenlong Zhao, Xiaolong Xu, Baokun Wang, Bowen Song, Changhua Meng, Tianyi Zhang, and Liang Chen. Sad: Semi-supervised anomaly detection on dynamic graphs. In Edith Elkind (ed.), Proceedings of the Thirty-Second International Joint Conference on Artificial Intelligence, IJCAI-23, pp. 2306–2314. International Joint Conferences on Artificial Intelligence Organization, 8 2023. doi: 10.24963/ijcai.2023/256.

---

> ### Author Response · Authors · 2025-11-26
> **W2 Clarification on Theoretical Foundation**
>
> ## W2 Clarification on Theoretical Foundation
> We sincerely appreciate the reviewer's feedback concerning the clarity of our theoretical foundation. We acknowledge that the original presentation was difficult to follow.
>
> To comprehensively address this concern, we have made substantial revisions to the manuscript: As detailed in our **Public Comment**, we have significantly enhanced the clarity of the theoretical section. In our newly uploaded version, we have introduced a **comprehensive notation table** in the Appendix A for easy reference; and we have added the **detailed derivations for key Equations (1), (3), and (7)** to ensure the mathematical rigor and improve the readability of our core formulation.
> We believe these major additions in the revised manuscript will fully address the concerns regarding the theoretical clarity.

---

> ### Author Response · Authors · 2025-11-26
> **W3 Typographical Issue (Dangling Lines)**
>
> ## W3 Typographical Issue (Dangling Lines)
> We thank the reviewer for pointing out this typographical issue regarding dangling lines (or widows/orphans).
>
> We agree that such formatting inconsistencies detract from readability. We have thoroughly reviewed the manuscript and revised the paragraph layouts to eliminate all instances of dangling lines that contain only a single word on the last line.
>
> We appreciate the attention to detail that helps us improve the overall professional presentation of the paper.

---

> ### Author Response · Authors · 2025-11-26
> **Q1 Clarification of the Results on Pesudo-Labels and Dynamic Labels**
>
> ## Q1 Clarification of the Results on Pesudo-Labels and Dynamic Labels
> We thank the reviewer for this question.
>
> As shown in Section 4.3, training with pseudo-labels consistently outperforms training with the original dynamic labels, despite the latter being the **ground-truth**. This counterintuitive result has also been observed in prior work such as SAD [1]. In Section 5.3 of SAD, the authors state:
> >“The result indicates that there is a lot of redundant or noisy information in the current graph datasets, which easily leads to overfitting of the model training. By dropping this part of label information and using pseudo-labeled data to participate in the training, the performance of the model is improved instead.”
>
> And our analysis in Section 4.3.2 provides a data-driven explanation: dynamic labels exhibit abrupt and noisy changes (consistency $C _ {u'} \equiv 0$), while our pseudo-labels enforce moderate temporal consistency, yielding smoother transitions and better feature alignment across time. In other words, the pseudo-labels act as a **temporal regularizer**, reducing inconsistencies in the original dynamic labels, which explains why they can outperform “true” dynamic labels in practice.
>
> ### **Reference**
> [1] Sheng Tian, Jihai Dong, Jintang Li, Wenlong Zhao, Xiaolong Xu, Baokun Wang, Bowen Song, Changhua Meng, Tianyi Zhang, and Liang Chen. Sad: Semi-supervised anomaly detection on dynamic graphs. In Edith Elkind (ed.), Proceedings of the Thirty-Second International Joint Conference on Artificial Intelligence, IJCAI-23, pp. 2306–2314. International Joint Conferences on Artificial Intelligence Organization, 8 2023. doi: 10.24963/ijcai.2023/256.  Main Track.

---

> ### Author Response · Authors · 2025-11-26
> **Q2 Comparison with ELI**
>
> ## Q2 Comparison with ELI
> We thank the reviewer for suggesting a comparison with ELI, a valuable work in the area of semi-supervised graph learning.
>
> We recognize ELI's importance, which is why we included a dedicated paragraph in our Related Work section (Apendix F.1 in the original version, G.1 in the revised version) to clearly delineate the two approaches. As detailed there, ELI is designed for **static graphs with globally sparse labels**, utilizes an **unsupervised label distribution estimation**, and incorporates labels via **graph-based regularization (Laplacian)**.
>
> In contrast, our work addresses the unique challenges of **dynamic graphs** where scarcity is **temporally driven** (labels only at the final timestamp). Our method uses a principled **Variational EM framework** for iterative refinement and a **Temporal Curriculum Learning** strategy for direct supervision. These core differences make the two methods conceptually distinct and address fundamentally different problem settings.
>
> Given that ELI is tailored for static graph scenarios, direct comparison on our dynamic graph benchmarks would require substantial modifications to the ELI framework to handle temporal dependency, which may not accurately reflect its original performance or design intention.
>
> Furthermore, in light of the computational demands associated with implementing the additional experiments requested by other reviewers, we unfortunately lack the necessary time and computational resources to furthur transfer ELI to dynamic graph setting, and implement and tune a modified ELI baseline for a fair comparison within the rebuttal period.
> We believe that the detailed comparison in our Related Work and the comprehensive experimental comparison against strong dynamic baselines (e.g., TGN, DyGFormer) fully validate our method's novelty and effectiveness in the intended dynamic graph domain.

---

> ### Author Response · Authors · 2025-11-26
> **Q3 Necessity of Our Pseudo-Labels and Contribution**
>
> ## Q3 Necessity of Our Pseudo-Labels and Contribution
> We sincerely thank the reviewer for this sharp and critical observation regarding the comparison between the Dynamic Label Supervision (DLS) baseline and the pseudo-label methods (CFT, NPL) in Table 1. This observation is crucial for clarifying our main contribution.
>
> ### **1. About DLS Performance and Label Quality**
> The reviewer is correct that the DLS baseline (which uses all available ground-truth dynamic labels) does not significantly improve over pseudo-label baselines and sometimes is even worse. In fact, this surprising result doesn't reduce our contribution, but addresses the rationality of our motivation instead:
> 1. **Extreme Consistency**: Our analysis of **Label Consistency** in **Section 4.3.2** shows that DLS supervision relies on labels that are often highly **inconsistent** across time steps (temporal consistency close to 0), as they represent potentially noisy, instantaneous state observations. Conversely, methods like CFT/NPL often use labels derived from aggregated or final information, leading to labels that are too rigid or **consistent** (consistency close to 1 or 0, but always extreme). Both extremes hinder effective temporal modeling.
> 2. **The Necessity of Our Pseudo-Labels**: Our Variational EM framework is designed to generate pseudo-labels that possess a varying and optimal level of temporal consistency (between 0 and 1). This varying consistency serves a critical purpose: it allows the model to learn **smooth temporal transitions** and facilitates better alignment of node features across different timestamps by bridging the gap between instantaneous noise and the stable final label. Therefore, the goal is not merely to recover the original dynamic labels, but to generate a superior, cleaner, and temporally stabilized supervision signal.
>
> ### **2. Contribution Reaffirmation (Beyond Curriculum Learning)**
> The success of the curriculum learning strategy is dependent on the quality of the signal it receives.
> 1. **Evidence from Experiment**: As shown in Section 4.3.1, using our generated pseudo-labels for direct supervision (without the curriculum strategy, denoted as PLS) already yields stronger performance than the DLS baseline. This empirically proves that the Variational Expectation-Maximization (VEM) framework successfully creates a learning signal that is qualitatively superior to the ground-truth dynamic labels in the dataset.
> 2. **Combined Contribution**: Our core contribution is the combination of the VEM framework for high-quality, temporally consistent pseudo-label generation and the curriculum learning strategy which then optimally utilizes these generated signals by prioritizing easier-to-classify examples first. Both components are necessary for the final superior performance.
> We have highlighted this distinction between simply recovering labels and generating a superior supervision signal in the discussion of **Section 4.3.2** of the revised paper.

---

### Official Review · Reviewer_NgF5 · 2025-10-29

**Soundness:** 2
**Presentation:** 2
**Contribution:** 2
**Rating:** 4
**Confidence:** 3

**Summary:**

This paper proposes a variational EM-based approach for label-limited dynamic node classification. Specifically, the authors treat each node’s label at the final timestamp as an observable variable, while the labels at historical interaction times are modeled as latent variables. They then employ an EM-based framework to learn the node classification model. In addition, the authors weight the pseudo labels generated from the variational posterior distribution based on the temporal gap between the final timestamp and the interaction time, aiming to mitigate the impact of unreliable pseudo labels.

**Strengths:**

- The problem of label-limited dynamic node classification is both practical and novel. Despite its relevance to real-world scenarios, it has received limited attention in prior work.
- Generally, the idea of adopting EM to solve this problem makes sense.

**Weaknesses:**

- The description of the method is ambiguous, which makes it difficult to evaluate. For example, when predicting node labels, the adopted dynamic graph model does not seem to consider the historical labels of nodes. It remains unclear how $p_{\theta}(y_u^t|G,Y_{F,B})$ is actually computed and how the model incorporates the historical labels $Y_{F,B}$.
- In the E-step, the variational loss is not actually utilized. The authors simply train the model using ground-truth labels, which breaks the connection with the EM framework.

**Questions:**

Please see the weaknesses.

---

> ### Author Response · Authors · 2025-11-26
> **W1: Clarification on Conditional Probability and Historical Label Incorporation**
>
> ## W1: Clarification on Conditional Probability and Historical Label Incorporation
> We sincerely thank the reviewer for this highly insightful and detailed feedback, particularly for pointing out the notational oversight and requesting a deeper clarification on the computational mechanism.
> ### **1. Clarity and Derivation**
> Regarding the concern that the description was unclear, we refer the reviewer to the **Appendix A & B** in our revised version (as detailed in our Public Comment). We have added a **comprehensive notation table** and included the **step-by-step derivation for some key equations**, including those related to the E-step, which should significantly improve followability.
> ### **2. Notational Correction**
> The reviewer is correct. We apologize for the typo in the conditional probability term. In fact, the correct expression should be conditional on all estimated historical labels except the label of the current node $u$ at time $t$. We have corrected the term $\hat{\mathcal{Y}} _ {E,B} \setminus \hat{\mathcal{Y}} _ {E,B}^u$ to $\hat{\mathcal{Y}} _ {E,B} \setminus \{\hat{y} _ u^t\}$ throughout the revised version.
> ### 3. Incorporating Historical Labels and Probability Computation (The Bridge between Theory and Practice):
> The reviewer raises a crucial question: how $P(y_u^t | \mathcal{G}, \hat{\mathcal{Y}} _ {E,B} \setminus \{\hat{y} _ u^t\})$ is actually computed by the model. This concerns the integration of the theoretical variational EM framework with practical GNNs.
> We follow the precedent set by semi-supervised graph learning works like GLEM [1] and GMNN [2]. In these approaches,GNN is leveraged as a direct model for conditional probability distribution.
>
> Critically, the dependence of the conditional probability on the labels of neighboring nodes ($\hat{\mathcal{Y}}_{E,B} \setminus \{\hat{y}_u^t\}$) is **not** introduced by explicitly passing these pseudo-labels as input features in the neural network's forward pass. Instead, this influence is **implicitly modeled** through the structure of the GNN itself. The GNN's message-passing mechanism—which aggregates information over the graph $\mathcal{G}$—is implicitly designed and trained (via the M-step objective) to learn the relationships that govern this conditional distribution, thus serving as the necessary bridge between the variational EM theory part and the GNN architecture.
>
> This integration provides the necessary **bridge between the theoretical framework Variational Expectation-Maximization and the practical deep learning architecture (GNN)**, allowing us to leverage the message-passing mechanism to effectively incorporate the influence of neighboring historical labels.
>
> [1] Jianan Zhao, Meng Qu, Chaozhuo Li, Hao Yan, Qian Liu, Rui Li, Xing Xie, and Jian Tang. Learning on large-scale text-attributed graphs via variational inference. ArXiv, abs/2210.14709, 2022.
>
> [2] Meng Qu, Yoshua Bengio, and Jian Tang. Gmnn: Graph markov neural networks. ArXiv, abs/1905.06214, 2019.

---

> ### Author Response · Authors · 2025-11-26
> **W2: E-step Implementation and Connection to the EM Framework**
>
> ## W2: E-step Implementation and Connection to the EM Framework
> We thank the reviewer for this astute observation regarding the practical implementation of the objective function, which clarifies a point not sufficiently detailed in the previous manuscript version. We recognize that this concern is fundamentally related to the role of pseudo-labels, which was also raised by Reviewer Ds7e (W3-Q3).
> ### **The Role of Pseudo-Labels and the VEM Framework**
> The reviewer is correct that the pseudo-labels are **exclusively utilized in the M-step** of our Variational Expectation-Maximization (VEM) framework. Our empirical finding in Section 4.2—where **PTCL ($\alpha \approx 0$) outperforms SEM ($\alpha \ne 0$)**—is a key practical result.
>
> This finding **does not negate the relevance of the VEM framework**. Instead, it highlights a crucial trade-off inherent in applying VEM to complex deep learning models with noisy latent variable estimations:
> 1. **Noise and Error Propagation**: Pseudo-labels generated for the historical data ($\mathcal{Y} _ {E,B}$) are inherently noisy estimations of the true, unobserved historical labels. Since the **decoder** is responsible for the final prediction (as shown in Algorithm 1), if its training objective (in the M-step) relies too heavily on these noisy pseudo-labels, it risks **significant error propagation**, which destabilizes training and compromises final accuracy.
> 2. **Prioritizing Ground Truth for Robustness**: To counter this, we **prioritize alignment with the high-quality, scarce ground-truth labels ($\mathcal{Y} _ {F,B}$)** by setting a low weight (small $\alpha$) for the pseudo-label term in the M-step. This modification ensures that the M-step loss function remains robust and anchored to the true labels.
> 3. **The Role of the Backbone**: The **dynamic graph backbone** (used to generate intermediate representations) is separate from the decoder's final prediction mechanism. This distinction implies that we need to constrain the training objective of the final, prediction-focused **decoder** much more tightly in the M-step than we might otherwise.
>
> In essence, we **retain the required iterative structure of VEM** (our core contribution, which addresses the latent variable problem) but implement a **modified, robust M-step loss** to achieve superior empirical performance and stability in the presence of noisy pseudo-labels. This demonstrates a strategic engineering decision that prioritizes effectiveness over strict mathematical adherence to the standard VEM M-step.
>
> We have **incorporated this detailed analysis and discussion** of the PTCL vs. SEM results into the revised **Section 4.2** to fully address this observation.

---

### Official Review · Reviewer_Ds7e · 2025-11-02

**Soundness:** 2
**Presentation:** 2
**Contribution:** 1
**Rating:** 0
**Confidence:** 4

**Summary:**

The authors of this work address a dynamic node classification task in temporal graphs, with the additional assumption that there are limited node labels available during training. The proposed approach builds on semi-supervised learning with pseudo-labeling, where the model is trained using data on a final time stamp to predict pseudo-labels for earlier timestamps. A curriculum learning approach is proposed, which progressively trains the model using (pseudo-)labels from later to earlier timestamps and which can use different GNN backbone learning techniques to generate node embeddings.

The proposed approach is evaluated in four different data sets, addressing binary node classification in data from two online platforms (binary target is to predict blocked users) and one financial social network (binary target is to predict users who do not pay back loans), as well as research areas in a scientific collaboration and citation data set. The proposed method shows mild performance improvements compared to baseline training techniques across all backbone GNN architectures. Further evaluation in an ablation study suggest a rather small contribution of the pseudo-labels and curriculum learning approach.

**Strengths:**

[S1] The paper addresses node classification in continuous-time temporal graphs (CTTG), which is an important and current topic in graph learning.

**Weaknesses:**

[W1] I could not follow the motivation for the very specific learning task considered in the paper, which assumes that node labels are available for a final but not for earlier timestamps. Moreover, some of the data sets used to motivate this do not seem to fit this setting, and I would argue that the problem should not be addressed from a dynamic node classification problem in the first place. See my questions Q1 below. Clarifying this is crucial, as the authors argue that a key contribution of their work is that they are the first to study this problem systematically.

[W2] Some key information is missing in the main manuscript (e.g. what are the actual labels in some of the data sets). Discussing the semantics of this in the main text is crucial to judge whether the data fits the proposed setting, see detailed comments in Q7 and Q8 below.

[W3] In found that some parts of the methdology, e.g. the application of the Variational Expectation Maximixation, are hard to follow and lack intuition. Also notation could be improved to make the method easier to understand, see my questions Q2, Q3, and Q6 and suggestions below.

[W4] While the proposed method shows a moderate improvement of average accuracy, the increase is small and even maximally simple baselines that essentially assume that labels are static (cf. W1) perform well. For some results, standard deviations are missing, which makes it impossible to judge whether the small observed differences are actually significant. See Q4 and Q5 below,

[W5] The contextualization of the method in terms of related work is very weak and important published works addressing the same problem even in the same data are neither mentioned nor used as a baseline, see Q9 below.

**Questions:**

[Q1] I could not follow why the Open Academic Graph data set motivates your setting of a dynamic graph, where "final static" node labels are available. First of all, I would argue that there are either static labels (that do not change over time) or final labels (indicating that they change but we only know their final values). More specifically, for the Open Academic Graph data set, research interest labels are rather aggregate labels, which result from a topic modelling method applied to all publications and venues of a researcher. I thus think that this does not fit the setting of the paper. Similarly, for the fraud setting, one could argue that the fraud label is rather an aggregate rather than a final label. Could the authors clarify why these settings correspond to a dynamic node classification problem (where labels are absent for earlier timestamps) instead of a simpler task where static classes are assigned to node based on a time series?

[Q2] I could not follow the description of the Expectation-Maximization approach. First of all, it is unclear to me what enters in the conditional probability. The notation seems to suggest that Y_{F, B} is only conditional on the dynamic graph G, which does not include any node labels. Is that correct? I think that section 3.1 should be explained better and I also believe that a better notation could help to make it easier to understand the approach.

[Q3] On page 4, the authors mention that, while there is a hyperparameter $\alpha$ that balances the influence of pseudo-labels and final labels in the objective function of the decoder, in practice a value of alpha works best, which actually suggests that pseudo-labels are unncessary (and which seems to support my criticism of the actual task relevance in Q1). Did I get this wrong? Are pseudo-labels only used in the maximization step? Could the authors clarify this?

[Q4] While the proposed approach shows the best average performance in most of the data sets, a few aspects of the results should be noted. First, fitting my earlier comments about the relevance of the task, two baselines which simply copy the final labels to all previous timestamps or only use available labels already perform well. Second, for all of the results, improvements are within the standard deviation intervals of other methods, which challenges the authors statement that their method significantly outperforms other methods.

[Q5] Related to my question Q3, the analysis in section 4.3 actually shows that the influence of pseudo-labels is - at best - small. Also, standard deviations are missing in the results, so it is impossible to judge whether the differences between the approach with and without pseudo-labels are actually significant. The same is true for table 3, which evaluates the contribution of the curriculum learning approach. Please add standard deviations

[Q6] Could the authors give an intuition what they want to capture with the consistency measure in equations 11 and 12?

[Q7] In the main text, I could not find information on what (binary) classification task is actually addresses for the Wikipedia, Reddit and Dsub data in the experiment evaluation. Additional information on page 18 of the appendix suggests that the binary labels refer to fraudulent behavior (DSub) or banned users (Wikipedia, Reddit). Please clarify this in the main text and include information on the label distribution and label dynamics.

[Q8] Regarding the fraud label for the DSub data set, I would first argue that the fact that users could not repay their loan is a default rather than a fraud, so please check your terminology. I think this semantics is crucial, as it also affects the classification task. In particular, I would argue that treating the question whether a credit customer will eventually default is not a reasonable setting for a dynamic node label prediction, as it is unlikely that a user will default multiple times. So, referring to Q1, to me this again seems to rather correspond to a simple (static) node classification problem in a dynamic graph.

[Q9] In the (overly short) related work section on dynamic node classification, the authors state that dynamic node classification remains underexplored, which is hardly the case. Many works in temporal graph learning have addressed different variants of the problem, e.g. assuming static node labels but using the temporal graph to classify them or using temporal embeddings to predict dynamic node labels. This is true for GNN-based methods like TGN, TGAT, DBGNN, TGBASE, DyREP, etc.

It is also not true that prior methods assume access to full dynamic labels, as some of these methods have considered semi-supervised or few-shot setting. In particular, fitting the setting considered by the authors and even building on a pseudo-label based training, the following method has been published two years ago:

S Tian et al.: SAD: Semi-Supervised Anomaly Detection on Dynamic Graphs, IJCAI 2023

The fact that this paper is not referenced and that the approach proposed in there is not considered as a baseline is a major omission.

Further suggestions:

- As a general rule, please define all abbreviations upon first use (EM for expectation maximization, OAG for Open Academic Graph, SSL for semi-supervised learning)

---

> ### Author Response · Authors · 2025-11-26
> **W1-Q1 Dataset Clarification and Motivation Explanation**
>
> ## W1-Q1 Dataset Clarification and Motivation Explanation
> We appreciate the reviewer's careful reading and valuable feedback. We would like to clarify the dynamic nature of our **CoOAG** dataset and the motivation behind our **label-limited setting**.
>
> While the **Open Academic Graph (OAG)** provides **static** research interest labels, our **CoOAG** dataset is fundamentally different in its **temporal dynamics**. In CoOAG, research interest labels are **not aggregate labels** derived from an author's entire publication history. Instead, as detailed in Appendix D.2.2, we generate **time-specific labels** using the Qwen-Plus API with a carefully designed prompt template that analyzes **only an author's most recent 10 papers at each timestamp**. This creates a genuine **dynamic labeling scenario** where research interests evolve over time as new publications emerge.
>
> The dynamic nature of CoOAG is substantiated by our statistics: **43.2%** of labeled nodes experience at least one label transition during their publication lifetime, with an average of **1.19** label changes per node. In Appendix D.2.3, we provide a concrete example (Node ID: 6816) showing a researcher who undergoes **three distinct label transitions** over a 10-year period, with clear shifts between *Robotics* and *Data Mining/Web Search* fields. These transitions reflect meaningful academic evolution rather than static aggregate interests.
>
> In practical scenarios like academic interest tracking, obtaining **complete historical labels** is prohibitively expensive. In CoOAG, labels are generated using a **large language model** (which could be replaced by human experts in other contexts). Obtaining labels for every timestamp would require repeatedly prompting the LLM or consulting experts about an author's research focus at multiple historical points—a process that multiplies annotation costs by the number of timestamps.
>
> Our method introduces **no significant additional parameters** beyond those of the underlying dynamic graph backbone. For reference, the parameter counts of our backbone models are modest:
>
> * **TGN** [1]: 0.86M
> * **TCL** [2]: 0.45M
> * **GraphMixer** [3]: 0.14M
> * **TGAT** [4]: 0.47M
> * **DyGFormer** [5]: 0.58M
>
> In contrast, the **Qwen-Plus** API used for generating our dataset labels is estimated to have **at least 70 billion parameters**, representing a parameter-efficiency improvement of **5–6 orders of magnitude** compared to LLM-based approaches.
>
> Our method addresses the challenge by requiring **only final-timestamp labels**, while still effectively capturing temporal dynamics through **pseudo-labeling** and our **Temporal Curriculum Learning** strategy. This approach significantly reduces annotation burden while achieving superior performance compared to methods that require complete dynamic labels.
>
> For the **DSub** (a subset of the DGraph dataset) setting, although each user ultimately receives a single fraud/non-fraud label, this label reflects a **latent behavioral state** that evolves over time and can be inferred from the temporal interaction patterns in the dynamic graph. Importantly, the **DGraph** [6] authors emphasize that dynamic graph modeling is crucial for detecting such anomalous behaviors. As stated in their paper, TGAT achieves the best performance because it:
>
> > "captures the most range of information, including dynamic information, node features and graph information."
>
> ### **References**
>
> [1] Emanuele Rossi, Ben Chamberlain, Fabrizio Frasca, Davide Eynard, Federico Monti, and Michael Bronstein. *Temporal graph networks for deep learning on dynamic graphs*. arXiv preprint arXiv:2006.10637, 2020.
>
> [2] Lu Wang, Xiaofu Chang, Shuang Li, Yunfei Chu, Hui Li, Wei Zhang, Xiaofeng He, Le Song, Jingren Zhou, and Hongxia Yang. *TCL: Transformer-based dynamic graph modelling via contrastive learning*. arXiv:2105.07944, 2021.
>
> [3] eilin Cong, Si Zhang, Jian Kang, Baichuan Yuan, Hao Wu, Xin Zhou, Hanghang Tong, and Mehrdad Mahdavi. *Do we really need complicated model architectures for temporal networks?* arXiv:2302.11636, 2023.
>
> [4] Da Xu, Chuanwei Ruan, Evren Korpeoglu, Sushant Kumar, and Kannan Achan. *Inductive representation learning on temporal graphs*. arXiv:2002.07962, 2020.
>
> [5] Le Yu, Leilei Sun, Bowen Du, and Weifeng Lv. *Towards better dynamic graph learning: New architecture and unified library*. NeurIPS 2023.
>
> [6] Xuanwen Huang, Yang Yang, Yang Wang, Chunping Wang, Zhisheng Zhang, Jiarong Xu, Lei Chen, and Michalis Vazirgiannis. *DGraph: A large-scale financial dataset for graph anomaly detection*. NeurIPS 2022.

---

> ### Author Response · Authors · 2025-11-26
> **W2-Q7 More Dataset Information Provided**
>
> ## W2-Q7 More Dataset Information Provided
> We sincerely appreciate the reviewer's valuable feedback regarding the missing information about dataset semantics and label definitions in the main text.
>
> In our revision, we incorporated the essential details directly into Section 4.1.1 (Datasets) of the main text.

---

> ### Author Response · Authors · 2025-11-26
> **W2-Q8 More Clarification about Dsub Dataset**
>
> ## W2-Q8 More Clarification about Dsub Dataset
> 1. **DSub is a subset of the DGraph dataset suite** [1]. We follow the terminology used by the dataset authors. As stated in Section 3.2 of the DGraph paper:
>
> > "We define users who exhibit at least one fraud activity, which means they do not repay the loans a long time after the due date and ignore the platform’s repeated reminders, as anomalies/fraudsters."
>
>    Our use of the term *fraud* is therefore directly inherited from the original dataset definition.
>
> 2. Although a user may not default multiple times, the **dynamic information in the graph remains essential**. As noted by the DGraph authors:
>
> > “Among all compared methods, TGAT achieves the state-of-the-art performance since it can capture the most range of information, including dynamic information, node features and graph information.”
>
>    Referring to Q1, this observation motivates us to model **DSub using dynamic graph methods**, which can capture temporal interaction patterns and potentially reveal the **latent behavioral states** of users over time—rather than treating the problem as assigning static labels to nodes based on a time series.
>
> ### **Reference**
>
> [1] Xuanwen Huang, Yang Yang, Yang Wang, Chunping Wang, Zhisheng Zhang, Jiarong Xu, Lei Chen, and Michalis Vazirgiannis. *DGraph: A large-scale financial dataset for graph anomaly detection*. NeurIPS 2022.

---

> ### Author Response · Authors · 2025-11-26
> **W3 Clarification on Expectation-Maximization Approach and Conditional Probability**
>
> ## W3 Clarification on Expectation-Maximization Approach and Conditional Probability
> We appreciate the reviewer's candid feedback that the application of the variational EM framework was "hard to follow and lacked intuition." This core component is critical to our methodology, and we have taken concrete steps to address both concerns.
>
> ### **1.  Regarding Clarity ("Hard to Follow")**
>
> We refer the reviewer to our Public Comment and the revised Appendix A & B, where we have provided a comprehensive notation table and the detailed, step-by-step derivations for key equations (Equations (1), (3), and (7)). We are confident that these additions will significantly improve mathematical clarity and allow the approach to be followed much more easily.
>
> ### **2. Regarding Intuition and Motivation ("Lack of Intuition")**
>
> The design choice of utilizing variational EM is directly motivated by the label-limited nature of our task: In our problem setting of dynamic node classification, the final labels $\mathcal{Y}_F$ are observed, but the historical node labels $\mathcal{Y}_E$ are unobserved due to annotation costs. These missing historical labels naturally correspond to the latent variables in the Variational EM framework. Therefore, variational EM algorithm is a canonical, principled approach for maximizing the likelihood when the data includes such latent variables. It allows us to iteratively:
>
> * **(E-step):** Estimate the posterior distribution of the unobserved historical labels (i.e., generate pseudo-labels).
> * **(M-step):** Optimise the model parameters based on both the observed final labels and the estimated historical pseudo-labels.
>
> Moreover, the application of variational EM is well-established in some related graph literature. Our approach is consistent with successful semi-supervised methods on static graphs, such as **GLEM** [1] and **GMNN** [2], which leverage the variational EM algorithm to iteratively refine predictions on unlabeled nodes. Our work logically extends this principled framework to the challenging label-limited, temporal setting of dynamic graphs.
>
> ### **References**
>
> [1] Jianan Zhao, Meng Qu, Chaozhuo Li, Hao Yan, Qian Liu, Rui Li, Xing Xie, and Jian Tang. *Learning on large-scale text-attributed graphs via variational inference.* ArXiv, abs/2210.14709, 2022.
>
> [2] Meng Qu, Yoshua Bengio, and Jian Tang. *GMNN: Graph Markov Neural Networks.* ArXiv, abs/1905.06214, 2019.

---

> ### Author Response · Authors · 2025-11-26
> **W3-Q2 Clarification on Expectation-Maximization Approach and Conditional Probability**
>
> ## W3-Q2 Clarification on Expectation-Maximization Approach and Conditional Probability
> We sincerely thank the reviewer for the careful reading and insightful comments on the clarity of our Expectation-Maximization (EM) formulation. We fully agree that improving the explanation of this core component is crucial.
>
> ### **1. Regarding the Conditional Probability**
>
> The reviewer’s understanding is correct. The conditional probability
> $ p(\mathcal{Y} _ {F, B} \mid \mathcal{G}) $
> is indeed conditioned solely on the dynamic graph  $\mathcal{G}$, which only comprises the node interactions and their timestamps, **without incorporating any node labels**.
>
> This formulation is central to our task for two main reasons:
>
> 1. **Standard Dynamic Graph Definition:**
>    As defined in Section 2 (Problem Formulation),
>    $
>    \mathcal{G} = { x(t_i) } = { (u_i, v_i, t_i) }
>    $,
>    this definition is consistent with prevalent dynamic graph learning literature, such as **TGN** [1] and **DyGFormer** [2].
>
> 2. **Semi-Supervised Objective:**
>    Our task is label-limited dynamic node classification, where the only ground-truth labels available for training are the final timestamp labels ( $ \mathcal{Y} _ {F, B} $ ), while the historical labels ( $ \mathcal{Y} _ {E, B} $ ) are considered unknown (latent variables).
>
>    Therefore, our training objective is to maximize the log-likelihood of the observed data, which includes the graph structure ( $ \mathcal{G} $ ) and the observed labels ( $ \mathcal{Y} _ {F, B} $ ).
>    The variational EM framework is thus naturally adopted to maximize
>    $
>    \log p(\mathcal{Y}  _ {F, B} \mid \mathcal{G})
>    $
>    by iteratively estimating the latent historical labels ( $\mathcal{Y} _ {E, B} $ ) through pseudo-labeling.
>
> ### **2. Regarding Clarity and Notation**
>
> We appreciate the suggestion for more succinct notation. While we have retained the current notation to maintain consistency with the field’s standard EM applications and to clearly distinguish between the final $ \mathcal{Y} _ F $ and historical $ \mathcal{Y} _ E $ label sets, we have taken a major step to improve overall clarity: **we have added a detailed notation table in Appendix A** (as outlined in our Public Comment) to serve as a comprehensive reference guide for all symbols used in the theoretical sections, including Section 3.1.
>
> ### **References**
>
> [1] Emanuele Rossi, Ben Chamberlain, Fabrizio Frasca, Davide Eynard, Federico Monti, and Michael Bronstein. *Temporal graph networks for deep learning on dynamic graphs.* arXiv preprint arXiv:2006.10637, 2020.
>
> [2] Le Yu, Leilei Sun, Bowen Du, and Weifeng Lv. *Towards better dynamic graph learning: New architecture and unified library.* Advances in Neural Information Processing Systems, 36:67686–67700, 2023.

---

> ### Author Response · Authors · 2025-11-26
> **W3-Q3: Role of Pseudo-Labels and $ \alpha $ Hyperparameter**
>
> ## W3-Q3: Role of Pseudo-Labels and $ \alpha $ Hyperparameter
> We thank the reviewer for this observation regarding the practical implementation of the objective function in the E-step. We apologise that this critical detail was not sufficiently analysed in the previous version of Section 4.2.
>
> ### **1. Clarification**
> The reviewer is correct: the pseudo-labels are used only in the M-step of the variational EM framework. Our empirical results in Section 4.2 shows that PTCL $ (\alpha = 0 $) performs better than SEM ( $ \alpha \neq 0$ ), but it does not negate the relevance of the VEM(Variational Expectation-Maximization) framework. Instead, it highlights a crucial trade-off between strict mathematical adherence to the standard variational EM M-step and engineering effectiveness in dealing with noisy pseudo-labels:
>
> 1. Pseudo-labels are inherently noisy estimations of the true historical labels ($ \mathcal{Y} _ {E,B} $). Since we use the decoder to predict the labels finally (Algorithm 1), if the decoder relies too heavily on them, it risks significant error propagation, which can destabilize training and reduce accuracy.
>
> 2. However, the dynamic graph backbone is only used to generate intermediate representations, so we don't need to constrain its training objective in M-step as tightly as we do to the decoder in E-step. By assigning a low weight to the pseudo-label term, we prioritize alignment with the high-quality, scarce ground-truth labels ( $ \mathcal{Y} _ {F,B} $ ). This allows the principled variational EM framework to guide the learning process and estimate the latent variables, while simultaneously mitigating the risk of the M-step loss function being corrupted by noisy predictions.
>
> ### **2. Conclusion**
> In essence, we retain the required iterative structure of variational EM (our core contribution) but implement a modified M-step loss for superior and more robust empirical performance. This demonstrates an important trade-off between strict mathematical adherence to the standard variational EM M-step and engineering effectiveness in dealing with noisy pseudo-labels. We have added this analysis and discussion of the PTCL vs. SEM results to the revised Section 4.2 to fully address this point.

---

> ### Author Response · Authors · 2025-11-26
> **W3-Q6 Explanation of Consistency Measure**
>
> ## W3-Q6 Explanation of Consistency Measure
> Thank you for the question. The consistency measure in Equations (11) and (12) is designed to quantify **how long a node’s dynamic labels remain the same with the final timestamp**. Intuitively:
> - $ \hat{N} _ {u'} $ measures the length of the longest consecutive suffix in which all intermediate labels are identical to the final label.
> - $ C_{u'} $ is the normalized consistency score, obtained by dividing $ \hat{N} _ {u'} $ by the total number of intervals, resulting in a value within ([0,1]).
> - For example, for a dynamic label sequence ([0,0,0,1,1,1]), we have $ \hat{N} _ {u'} = 2 $ and $ C _ {u'} = 0.4 $；for another sequence ([0,0,0,0,0,1]), we have $ \hat{N} _ {u'} = 0 $ and $ C _ {u'} = 0 $.
>
> Thus, a high consistency score indicates **gradual and stable label evolution toward the final state**, while a low score indicates **abrupt or inconsistent label changes** over time. As shown in the paper, our method produces labels with moderate and realistic consistency, avoiding the abrupt changes of dynamic labels and the overly rigid continuity enforced by CFT, enabling smooth temporal transitions and better aligning features across time.

---

> ### Author Response · Authors · 2025-11-26
> **W4-Q4 Explanation of Performance Superiority**
>
> ## W4-Q4 Explanation of Performance Superiority
> We appreciate the reviewer's thoughtful comments regarding the performance gains and statistical significance of our results.
> Our method demonstrates significant and meaningful improvements over key baselines:
>
> PTCL consistently outperforms the Copy-Final Timestamp (CFT) baseline by considerable margins across almost all datasets and backbone architectures. For instance, with the TGAT backbone on Wikipedia, PTCL achieves 85.52% compared to CFT's 77.43%—**an 8.09 percentage point improvement**. Similarly, on the CoOAG dataset, PTCL improves upon CFT by **2.77 percentage points** with the same backbone. These improvements are not marginal but represent significant advances in capturing temporal dynamics that CFT completely ignores by assuming static labels throughout the timeline.
>
> Most remarkably, PTCL frequently exceeds the performance of Dynamic Label Supervision (DLS), which has access to complete dynamic label sequences during training. For example, with the TCL backbone on Reddit, PTCL achieves 89.41% compared to DLS's 82.85%—**a 6.56 percentage point improvement** despite using significantly less supervision (only final timestamp labels vs. complete historical labels). This demonstrates that PTCL effectively recovers the temporal evolution patterns that are critical for accurate classification, without requiring expensive intermediate label annotations.
>
> In the field of dynamic graph learning, consistent performance improvements across diverse datasets and model architectures are widely accepted as evidence of method effectiveness. Our approach demonstrates advantages in 18 out of 19 experimental settings, with substantial **average improvements of approximately 5%** over CFT. While dynamic graph learning inherently exhibits some performance variability across runs due to the sensitivity of temporal modeling to initialization and sampling patterns, the consistent direction and magnitude of improvements across multiple backbone architectures provide strong evidence of our method's effectiveness.
>
> For context, established improvements between foundational methods in this field (such as TGN over TGAT) typically report **average gains of  ~2.8%**. Our method's ability to achieve superior performance with only final timestamp labels—while often exceeding methods with full dynamic supervision—represents a significant advance for practical applications where complete label sequences are unavailable or prohibitively expensive to obtain.

---

> ### Author Response · Authors · 2025-11-26
> **W4-Q5 Influence of Pseudo-Labels and the Absence of Standard Deviations**
>
> ## W4-Q5 Influence of Pseudo-Labels and the Absence of Standard Deviations
> Thank you for the insightful comment. We address your concerns regarding the influence of pseudo-labels and the absence of standard deviations as follows.
>
> ### **1. The influence of pseudo-labels is meaningful and consistent.**
>
> Section 4.3 evaluates the role of pseudo-labels through a controlled study: we train each backbone using pseudo-labels (generated by our trained model) as full supervision, and compare the results with Dynamic Label Supervision (DLS). The results in the updated Table 2 show that models trained with our pseudo-labels consistently outperform those trained with the original dynamic labels, with an average AUC improvement of +0.96%, even though—in theory—using the ground-truth dynamic labels should be more advantageous.
>
> This phenomenon has also been observed in prior work such as the SAD[1], where using pseudo-labeled data surprisingly leads to performance gains [in Section 5.3]. And our analysis in Section 4.3.2 provides a data-driven explanation:
> - dynamic labels exhibit abrupt and noisy changes (consistency $ C _ {u'} \equiv 0 $),
> - while our pseudo-labels achieve moderate and realistic consistency, enabling smoother temporal transitions and better feature alignment across time.
>
> Thus, the benefit of pseudo-labels is not marginal noise. It reflects their ability to regularize temporal supervision and reduce inconsistency in the original dynamic labels.
>
> ### **2. Standard deviations have been added to all relevant tables.**
>
> Following your suggestion, we have updated Table 2 and Table 3 to include standard deviations across multiple runs.
>
> ### **Reference**
> [1] Sheng Tian, Jihai Dong, Jintang Li, Wenlong Zhao, Xiaolong Xu, Baokun Wang, Bowen Song, Changhua Meng, Tianyi Zhang, and Liang Chen. Sad: Semi-supervised anomaly detection on dynamic graphs. In Edith Elkind (ed.), Proceedings of the Thirty-Second International Joint Conference on Artificial Intelligence, IJCAI-23, pp. 2306–2314. International Joint Conferences on Artificial Intelligence Organization, 8 2023. doi: 10.24963/ijcai.2023/256.  Main Track.

---

> ### Author Response · Authors · 2025-11-26
> **W5-Q9-1 More Related Work**
>
> ## W5-Q9-1 More Related Work
> We thank the reviewer for the insightful comments. Following your suggestion, we have substantially expanded and clarified the related work section on temporal graph learning and dynamic node classification.
>
> First, we agree that dynamic node classification has been studied under some formulations, including settings where static labels are inferred using temporal structure or where temporal embeddings are used for timestamp-wise prediction. To reflect this breadth, our revised section now covers not only classical temporal GNNs such as TGN, TGAT, DyRep [1], DBGNN [2], and TGBase [3], but also more recent dynamic node classification methods including TADGNN [4] and OTGNet.
>
> We also appreciate your highlighting the omission of SAD [5]. We are currently evaluating SAD under our label-limited dynamic node classification setting, and we plan to include these results in the updated experimental section once the evaluation is complete.
>
> ### **References**
> [1] Rakshit Trivedi, Mehrdad Farajtabar, Prasenjeet Biswal, and Hongyuan Zha. Dyrep: Learning representations over dynamic graphs. In International Conference on Learning Representations (ICLR), 2019.
>
> [2] Lisi Qarkaxhija, Vincenzo Perri, and Ingo Scholtes. De bruijn goes neural: Causality-aware graph neural networks for time series data on dynamic graphs. In Bastian Rieck and Razvan Pascanu (eds.), Proceedings of the First Learning on Graphs Conference, volume 198 of Proceedings of Machine Learning Research, pp. 51:1–51:21. PMLR, 09–12 Dec 2022.
>
> [3] Farimah Poursafaei, Zeljko Zilic, and Reihaneh Rabbany. A Strong Node Classification Baseline for Temporal Graphs, pp. 648–656. doi: 10.1137/1.9781611977172.73.
>
> [4] Jiarui Sun, Mengting Gu, Chin-Chia Michael Yeh, Yujie Fan, Girish Chowdhary, and Wei Zhang. Dynamic graph node classification via time augmentation, 2022.
>
> [5] Sheng Tian, Jihai Dong, Jintang Li, Wenlong Zhao, Xiaolong Xu, Baokun Wang, Bowen Song, Changhua Meng, Tianyi Zhang, and Liang Chen. Sad: Semi-supervised anomaly detection on dynamic graphs. In Edith Elkind (ed.), Proceedings of the Thirty-Second International Joint Conference on Artificial Intelligence, IJCAI-23, pp. 2306–2314. International Joint Conferences on Artificial Intelligence Organization, 8 2023. doi: 10.24963/ijcai.2023/256. Main Track.

---

> ### Author Response · Authors · 2025-11-26
> **W5-Q9-2 Define all Abbreviations upon First Use**
>
> ## W5-Q9-2 Define all Abbreviations upon First Use
> We appreciate the reviewer’s comment. In response, we have standardized terminology throughout the paper by defining all abbreviations—such as EM (Expectation Maximization), OAG (Open Academic Graph), and SSL (Semi-Supervised Learning)—upon their first use.

---

### Author Response · Authors · 2025-11-26
**Summary of Changes in the Revised paper**

## Summary of Changes in the Revised Paper

We sincerely thank all reviewers for their insightful and constructive feedback. The suggestions have been very valuable and helpful in helping us improve the clarity, theoretical rigor, and experimental completeness of our work. Below is a summary of the major revisions incorporated into the new version of our paper:

### I. Theoretical Part

Some reviewers noted that the theoretical part was "hard to follow" and lacked detailed explanations. To address this fundamental concern, we have made the following significant additions, primarily in the Appendix:

* We have introduced a **dedicated notation table** in Appendix A to centralize some important mathematical symbols and their definitions, ensuring readers can easily reference the necessary context for each equation.
* We have added **detailed derivations** for Eq. (1), (3), and (7) in Appendix B to provide a transparent and rigorous explanation of the mathematical formulation of our framework.
* We have carefully proofread the entire paper and **corrected several minor typos** in the original formulas (e.g., change $\hat{\mathcal{Y}} _ {E,B} \setminus \hat{\mathcal{Y}} _ {E,B}^u$ to $\hat{\mathcal{Y}} _ {E,B}\setminus \{\hat{y} _ u^t\}$ in Eq. 3).

### II. Experiments Part
* We have **expanded dataset descriptions** in Section 4.1.1, particularly clarifying the binary classification tasks for Wikipedia (predicting temporarily banned users), Reddit (identifying banned users), and Dsub (fraud detection).
* We have **added analysis of our method and SEM** in Section 4.2.
* We have updated Table 2 and Table 3 to include standard deviations across multiple runs.
* We have moved the detailed explanation of FLiD from the appendix into the main text in Section 4.1.3.
* We are running experiments about a new baseline SAD and a new dataset Dsub-1M. Results will be updated as soon as possible.

### Others
* We have provided a more comprehensive related work section on dynamic graphs.
* We have **standardized terminology** throughout the paper by defining all abbreviations upon first use, including EM (Expectation Maximization), OAG (Open Academic Graph), and SSL (Semi-Supervised Learning), improving readability and accessibility.
* We have updated figure 3 to make it more clear to understand. Specifically, 'BB' represents backbone, and 'Dec' represents decoder. We also add caption of the red box standing for 'being joinytly optimized' to the figure.
* We have revised the paragraph layouts to eliminate all instances of dangling lines that contain only a single word on the last line.
* We have provided a more detailed description in Appendix D.2.2.

We are confident that these substantial revisions have improved the quality and accessibility of our paper and address the points raised by the reviewers.

---

### Note · Authors · 2025-12-01

I have read and agree with the venue's withdrawal policy on behalf of myself and my co-authors.